# No Prior, No Leakage: Revisiting Reconstruction Attacks in Trained Neural Networks

**Yehonatan Refael** *
Department of Electrical Engineering
Tel-Aviv University, Israel
refaelkalim@mail.tau.ac.il

**Guy Smorodinsky**\*
Faculty of Computer and Information Science
Ben-Gurion University of the Negev, Israel
smorodin@post.bgu.ac.il

**Ofir Lindenbaum** †
Faculty of Engineering
Bar-Ilan University, Israel
ofir.lindenbaum@biu.ac.il

**Itay Safran**†
Faculty of Computer and Information Science
Ben-Gurion University of the Negev, Israel
safrani@bgu.ac.il

## Abstract

The memorization of training data by neural networks raises pressing concerns for privacy and security. Recent work has shown that, under certain conditions, portions of the training set can be reconstructed directly from model parameters. Some of these methods exploit implicit bias toward margin maximization, suggesting that properties often regarded as beneficial for generalization may actually compromise privacy. Despite compelling empirical demonstrations, the reliability of these attacks remains poorly understood and lacks a solid theoretical foundation. In this work, we take a complementary perspective: rather than designing stronger attacks, we analyze the inherent weaknesses and limitations of existing reconstruction methods and identify conditions under which they fail. We rigorously prove that, without incorporating prior knowledge about the data, there exist infinitely many alternative solutions that may lie arbitrarily far from the true training set, rendering reconstruction fundamentally unreliable. Empirically, we further demonstrate that exact duplication of training examples occurs only by chance. Our results refine the theoretical understanding of when training-set leakage is possible and provide new insights into mitigating reconstruction attacks. Remarkably, we demonstrate that networks trained more extensively, and therefore satisfying implicit bias conditions more strongly, are, in fact, less susceptible to reconstruction attacks, reconciling privacy with the need for strong generalization in this setting.

## 1 Introduction

Neural networks have achieved remarkable success across a wide variety of tasks, but their use raises fundamental concerns of privacy (Abadi et al., 2016b; Runkel et al., 2024; Fang et al., 2024; Tan et al., 2024; Bombari & Mondelli, 2025). Recent work has demonstrated that portions of the training data can be reconstructed directly from the parameters of a trained model, even without access to gradients or queries (Haim et al., 2022; Buzaglo et al., 2023b; Loo et al., 2023). Unlike some previous methods that produce generic reconstructions resembling class prototypes or averages (Carlini et al., 2021; 2019), the new techniques generate highly accurate and specific reproductions of the original training data. Such reconstruction attacks highlight the risk that sensitive or private information may leak from models, undermining their safe deployment in practice.

Despite the alarming success of these attacks, our theoretical understanding of them remains limited. In particular, the attack introduced in (Haim et al., 2022), along with some subsequent work inspired by it, such as (Buzaglo et al., 2023b), builds on results concerning the implicit bias of gradient-based optimization in training homogeneous networks (Lyu & Li, 2020; Ji & Telgarsky, 2020).[1]

---

*Equal Contribution.
†Co-last authors.
[1]Specifically, we focus on the KKT conditions for binary classification as defined in Theorem 1.

Intuitively, when optimization succeeds under certain assumptions, such networks trained using standard techniques do not merely converge to any solution that fits the training set; instead, they converge to specific solutions that satisfy additional constraints. Building on this observation, Haim et al. (2022) constructs an objective function that is minimized when these constraints are satisfied and carries out their attack by optimizing it. Nevertheless, despite its theoretical foundations, our understanding of the conditions under which the attack succeeds remains rudimentary, as rigorous analyses of this setting are scarce.

Recently, Smorodinsky et al. (2025) provided rigorous guarantees on the efficacy of such a reconstruction attack. However, these guarantees rely on restrictive assumptions, such as a univariate data distribution, which may limit their practical applicability. This leaves many fundamental questions open and motivates several follow-up directions to establish clear conditions under which such attacks can be provably effective or provably mitigated. A central question that arises is:

> *To what extent do the constraints imposed by the implicit bias of trained neural networks leak information about the training data?*

In this paper, we address the above question by rigorously studying properties of the objective function used to fuel the reconstruction attack introduced by Haim et al. (2022). We propose a method that, given the original training data, enables the construction of multiple other global minima for the objective function used in the attack. Additionally, we demonstrate that under certain conditions, these global minima are ubiquitous, and that solutions to this optimization problem exist that are substantially different from the original training data.[2] Moreover, we show that the closer the network is to a solution, the harder it becomes to successfully execute such attacks, perhaps contrary to previous common wisdom. Lastly, we empirically demonstrate that if an attacker initializes the attack far from the true training data, the reconstructed instances are significantly farther from the training data as well. This highlights that additional *prior knowledge* (as used by Haim et al. (2022)) is crucial for successful reconstruction attacks in this setting, and that the constraints imposed by the implicit bias alone do not necessarily reveal information about the training data.

The rest of this paper is structured as follows: After specifying our contributions in detail below, we turn to discuss additional related work. In Section 2, we present the required background and notation used throughout this paper. In Section 3 we theoretically study the set of feasible solutions to the objective function used in implicit-bias-driven privacy attacks, and in Section 4 we empirically validate and support our theoretical findings. Lastly, Section 5 summarizes our contributions and discusses potential limitations and future work directions.

**Our contribution.** We prove that the attack presented by Haim et al. (2022) has inherent limitations. In particular, we demonstrate that prior knowledge about the data domain is necessary to accurately recover the true training examples, and we empirically show several scenarios in which these attacks fail. More specifically:

- In Subsection 3.1, we demonstrate that in the reconstruction attack devised by Haim et al. (2022), there are numerous potential candidate training sets that seem indistinguishable from each other. We provide simple, constructive techniques for generating alternative candidates, either by merging two data instances into a single instance (Lemma 2) or by splitting a single instance into two (Lemma 3). Moreover, we demonstrate that under the assumption that the training data does not span the entire domain, there exist infinitely many such alternative training sets whose distance from the true training set can be arbitrarily large (Theorem 4). Lastly, we relax this assumption, assuming the training data only approximately lies on a linear subspace, and we further analyze how far a point can be split (Theorem 5).

- In Subsection 3.2, we consider the more realistic setting in which the trained model only approximately satisfies the implicit bias constraints, and present merging and splitting techniques analogous to those mentioned in the previous bullet point (Lemmas 6 and 7). We then assume that the attacker has limited knowledge of the proximity to an implicit-bias

---

[2]It is noteworthy that this result answers a question which was raised in Haim et al. (2022): "On the theoretical side, it is not entirely clear why our optimization problem in Equation 5 converges to actual training samples, even though there is no guarantee that the solution is unique, especially when using no prior." – solutions are ubiquitous rather than unique, and in the absence of prior knowledge, the attack fails.

solution and analyze the extent to which individual points can be split (Theorem 8). Finally, we demonstrate that even if the attacker possesses additional information about the training procedure, under the assumptions of structured data and a well-trained model, it remains possible to split the points in a way that preserves data confidentiality (Theorem 9).

- In Section 4, we complement our theoretical results with experiments that support our findings. Specifically, we model the attacker's prior as knowledge of the data domain boundaries, incorporated into the reconstruction attack through the attacker's initialization. We demonstrate, on both synthetic data and CIFAR, that as the attacker's prior weakens, the effectiveness of the attack decreases accordingly, with convergence to solutions far from the true training set, as predicted by our theory.

## ADDITIONAL RELATED WORK

**Reconstruction-based privacy attacks.** An emerging line of work studies the possibility of extracting training data directly from trained models, raising serious privacy concerns. Such reconstruction attempts have been demonstrated across a range of settings, including large generative models (Carlini et al., 2019; 2021; Nasr et al., 2023), diffusion models and diffusion architectures (Somepalli et al., 2022; Carlini et al., 2023), and federated learning (Zhu et al., 2019; He et al., 2019; Hitaj et al., 2017; Geiping et al., 2020; Huang et al., 2021; Wen et al., 2022) scenarios. Several studies, most notably Haim et al. (2022); Buzaglo et al. (2023a), have proposed optimization-based strategies that leverage the implicit bias of neural networks to recover examples from the original training data. Their approach frames data recovery as the problem of minimizing a suitably defined objective, whose solution can align with portions of the true training set.

A complementary line of work, exemplified by Loo et al. (2023), studies attacks on fine-tuned models that reconstruct downstream private training data, in contrast to the setting studied by Haim et al. (2022), which we also examine in this paper. This scenario is particularly important, given the widespread use of fine-tuning on public models (e.g., LLaMA, DeepSeek, Mistral). Nonetheless, the theoretical guarantees presented by Loo et al. (2023) are established under simplified and somewhat unrealistic settings that do not fully capture practical conditions, as the underlying proof assumes a particular infinite-width network model and data that lie exactly on the unit hypersphere.

A different perspective investigates the reliability of reconstruction attacks. The study in Runkel et al. (2024) empirically shows that reconstruction is highly sensitive to initialization, leading to the generation of plausible samples not present in the original training data. This ambiguity makes it difficult for an adversary to verify whether a recovered image is an actual training sample. While their empirical findings are similar to ours, our work provides rigorous theoretical guarantees that underpin this phenomenon, whereas their study remains purely empirical.

**Differential Privacy.** The current "golden standard" for privacy is the differential privacy framework Dwork et al. (2006b); Dwork & Roth (2014). Intuitively, differential privacy quantifies how much a model's output can change when a single data instance is modified, thereby providing a worst-case guarantee over all neighboring datasets. The smaller this change, the stronger the privacy guarantee. In practice, mechanisms enforce differential privacy by injecting carefully calibrated noise into data, queries, or training procedures (Dwork et al., 2006a; Abadi et al., 2016a). The composition and accounting results quantify the cumulative privacy loss (Kairouz et al., 2015; Balle & Wang, 2018). These guarantees often come with utility trade-offs, which are studied in both ERM and deep learning (Chaudhuri et al., 2011; Abadi et al., 2016a). Unlike differential privacy, our approach does not rely on noise injection. Instead, we seek to better understand the underlying causes of implicit-bias-driven privacy vulnerabilities, so that they can be circumvented when possible.

## 2 PRELIMINARIES, BACKGROUND AND NOTATION

**Notation and terminology.** We consider binary classification with data $(\mathbf{x}, y) \in \mathbb{R}^d \times \{-1, 1\}$ and training set $\{(\mathbf{x}_i, y_i)\}_{i=1}^n$. Let $\Phi(\boldsymbol{\theta}; \cdot) : \mathbb{R}^d \to \mathbb{R}$ be a neural network with parameters $\boldsymbol{\theta} \in \mathbb{R}^k$. Write $[x]_+ := \max(0, x)$. A homogeneous 2-layer ReLU network is $\Phi(\boldsymbol{\theta}, \mathbf{x}) = \sum_{j=1}^k v_j \left[\mathbf{w}_j^\top \mathbf{x} + b_j\right]_+$, where $\boldsymbol{\theta} = (\mathbf{w}_j, v_j, b_j)_{j=1}^k$. The unit sphere is $\mathbb{S}^{d-1} := \{\mathbf{x} \in \mathbb{R}^d : \|\mathbf{x}\|_2 = 1\}$. Let $d(\mathbf{x}_1, \mathbf{x}_2)$ denote the distance between vectors (Euclidean by default), and for sets $A, B$ define $d(A, B) :=$

$\min_{a \in A, b \in B} d(a, b)$. For $\mathbf{x} \in \mathbb{R}^d$, its activation pattern is the binary vector whose $j$-th entry indicates whether neuron $j$ is active on $\mathbf{x}$. For neuron $(\mathbf{w}_j, b_j)$ and point $\mathbf{x}$, set $D_j(\mathbf{x}) := \frac{\langle \mathbf{w}_j, \mathbf{x} \rangle + b_j}{\|\mathbf{w}_j\|}$, the signed distance to the hyperplane $\langle \mathbf{w}_j, \mathbf{x} \rangle + b_j = 0$.

**Preliminaries.** The following well-established result, which has recently attracted significant attention, shows that homogeneous neural networks trained with logistic or exponential loss exhibit an implicit bias: they converge in direction to the solution of a max-margin problem. This implicit bias is the key phenomenon underlying the attacks we study in this paper.

**Theorem 1** (paraphrased version of Lyu & Li (2020), Ji & Telgarsky (2020)). *Let $\Phi(\boldsymbol{\theta}; \mathbf{x})$ be a normalized homogeneous[3] ReLU neural network. Consider minimizing the logistic ($z \mapsto \log(1 + e^{-z})$) or the exponential ($z \mapsto e^{-z}$) loss using gradient flow (which is a continuous time analog of gradient descent) over a binary classification set $\{(\mathbf{x}_i, y_i)\}_{i=1}^n \subseteq \mathbb{R}^d \times \{-1, 1\}$. Assume that there is a time $t_0$ where $L(\boldsymbol{\theta}(t_0)) < \frac{1}{n}$. Then, gradient flow converges in direction[4] to a first order stationary point (KKT point Vapnik (1995)) of the following maximum-margin problem:*

$$\min_{\boldsymbol{\theta}} \frac{1}{2} \|\boldsymbol{\theta}\|^2 \ \ s.t \ \ \forall i \in [n] \ \ y_i \Phi(\boldsymbol{\theta}; \mathbf{x}_i) \geq 1.$$

A KKT point of Equation 1 is characterized by the following set of conditions:

$$\boldsymbol{\theta} = \sum_{i=1}^n \lambda_i \nabla_{\boldsymbol{\theta}} [y_i \Phi(\boldsymbol{\theta}; \mathbf{x}_i)] \qquad \text{(stationarity)} \qquad (1)$$

$$y_i \Phi(\boldsymbol{\theta}; \mathbf{x}_i) \geq 1, \forall i \in [n] \qquad \text{(primal feasibility)} \qquad (2)$$

$$\lambda_i \geq 0, \forall i \in [n] \qquad \text{(dual feasibility)} \qquad (3)$$

$$\lambda_i = 0 \text{ if } y_i \Phi(\boldsymbol{\theta}; \mathbf{x}_i) \neq 1, \forall i \in [n] \qquad \text{(complementary slackness)} \qquad (4)$$

Utilizing the above result, Haim et al. (2022) devised the following reconstruction attack, demonstrating that one can reconstruct a substantial subset of the training data from a deployed model.

The core of the method is a minimization problem. It simultaneously adjusts the candidate training data $\mathbf{X}'$ and their associated Lagrange multipliers $\lambda$ to minimize a composite loss function. This loss function is designed to enforce several of the KKT conditions that characterize an optimal solution for the classifier. The objective function is given explicitly by

$$\mathbf{X}' = \arg \min_{\{\lambda_i, x_i', i \in [n]\}} \underbrace{\left\| \boldsymbol{\theta} - \sum_{i=1}^m \lambda_i \nabla_{\theta} [y_i \Phi(\theta; x_i')] \right\|}_{L_{\text{stationary}}} + \underbrace{\sum_{i=1}^m \max\{-\lambda_i, 0\}}_{L_{\lambda}} + L_{\text{prior}}(X'). \qquad (5)$$

The stationary loss in Equation 5 penalizes deviations from the KKT stationary condition (Equation 1), where the non-negativity loss enforces the non-negativity constraint on the multipliers $\lambda_i$, as required by Equation 3, and $L_{\text{prior}}$ represents some prior knowledge we might have about the dataset. Lastly, note that the margin's value, as defined by the primal feasibility condition in Equation 2 at a KKT point, is normalized (Lyu & Li, 2020) and thus equals 1. However, in practice, the margin width $\gamma(\boldsymbol{\theta})$ may take on a different scale, which, for later use, we denote by $\gamma(\boldsymbol{\theta}) = p > 0$.[5] We remark that this method aims to reconstruct training data instances that are on the margin (namely, instances whose prediction value is $p$ or $-p$), since otherwise, by Equation 4, they do not factor in the constructed objective.

Although incorporating prior knowledge as an additional loss term to $L_{\text{KKT}}$ is a common technique, we deliberately exclude it from our analysis. Our rationale is that such priors are often heuristic and highly context-specific, rather than fundamental to the reconstruction method in Equation 5. Including a prior would confound the evaluation of the method's performance, as success would become dependent on the specificity of the external knowledge, making it difficult to disentangle the contribution of the mathematical constraints imposed by Equations 1-4.

---

[3]A network $\Phi(\boldsymbol{\theta}; \mathbf{x})$ is called *homogeneous* if there exists $c > 0$ such that for every $b > 0$, $\boldsymbol{\theta}$ and $\mathbf{x}$, it holds that $\Phi(b \cdot \boldsymbol{\theta}; \mathbf{x}) = b^c \Phi(\boldsymbol{\theta}; \mathbf{x})$.

[4]We say that gradient flow *converges in direction* to $\hat{\boldsymbol{\theta}}$ if $\lim_{t \to \infty} \frac{\boldsymbol{\theta}(t)}{\|\boldsymbol{\theta}(t)\|} = \frac{\hat{\boldsymbol{\theta}}}{\|\hat{\boldsymbol{\theta}}\|}$.

[5]The margin $\gamma(\boldsymbol{\theta})$ scales linearly with $\|\boldsymbol{\theta}\|_2^c$, where $c$ is the order of homogeneity, namely $\tilde{\gamma}(\boldsymbol{\theta}) := \frac{\gamma(\boldsymbol{\theta})}{\|\boldsymbol{\theta}\|_2^c}$.

# 3 IMPLICIT-BIAS-BASED RECONSTRUCTION ATTACKS

In this section, we theoretically study the reconstruction attack introduced by Haim et al. (2022). We begin by formally defining our framework and the required definitions, which differ slightly from those in the practical setting studied in their seminal work.

**Framework.** An attacker, motivated to reconstruct the training set $S$, is given a full access to the neural network (binary classifier) architecture $\Phi(\boldsymbol{\theta}; \cdot)$, as well as complete knowledge of the weights $\boldsymbol{\theta}$.[6] However, the attacker has no knowledge of the training samples $\mathbf{X} = (\mathbf{x}_1, \dots, \mathbf{x}_n)$ or the quantity $n$. In addition, we first assume no knowledge of the margin scale $\gamma(\boldsymbol{\theta}) = p$ in the following subsection, and in Subsection 3.2 we relax this assumption. With this information, the attacker then optimizes the objective in Equation 5 to reconstruct the training set, yielding $\mathbf{X}' = (\mathbf{x}'_1, \dots, \mathbf{x}'_n)$.

We begin by defining the KKT loss, which is used to assess the feasibility of a reconstructed set. As mentioned in the previous section, this definition omits the prior term.

**Definition 1** (KKT-loss). *The* KKT-loss *is defined as,*

$$L_{\mathrm{KKT}}(\mathbf{x}_1, \dots, \mathbf{x}_l, \lambda_1, \dots, \lambda_l) := \gamma_1 L_{\mathrm{stationary}}(\mathbf{x}_1, \dots, \mathbf{x}_l, \lambda_1, \dots, \lambda_l) + \gamma_2 L_\lambda(\lambda_1, \dots, \lambda_l), \quad (6)$$

*where $\gamma_1, \gamma_2 > 0$ denote hyperparameters controlling the optimization process.*

**Remark 1.** *Note that $l$, the number of data samples in Definition 1, does not necessarily equal $n$. This is justified, as by assumption, the attacker does not know the number of training samples $n$. Additionally, we do not incorporate the constraints in Equation 2 and Equation 4 into the definition of $L_{\mathrm{KKT}}$, as these were also not used by Haim et al. (2022). A detailed discussion about this can be found in Appendix E.*

Following the definition of our loss function, we specify the corresponding set of examples that satisfy the conditions of Theorem 1, which the minimization of the objective in Equation 6 aims to find.

**Definition 2** (KKT set). *Let $\Phi(\boldsymbol{\theta}; \cdot)$ be a binary classification network with weights $\boldsymbol{\theta}$ that has converged to a KKT point (satisfying Equations 1–4). A set of inputs $S = \{\mathbf{x}_1, \dots, \mathbf{x}_l\}$ is called a* KKT *set if there exist nonnegative multipliers $\lambda_1, \dots, \lambda_l \geq 0$ such that $L_{\mathrm{KKT}}(S, \lambda_1, \dots, \lambda_l) = 0$.*

Clearly, the original training set $S$ is a KKT set by assumption.

## 3.1 CONVERGENCE TO AN EXACT KKT POINT

We now show that, perhaps surprisingly, once the prior component is removed from Equation 5, the objective $L_{\mathrm{KKT}}$ admits infinitely many global minima. Moreover, certain minima yield reconstructed samples that differ substantially from the original training set, as measured by the minimum Euclidean distance between neighboring instances. Consequently, the reconstruction attack cannot reliably distinguish the actual training set from these alternative KKT sets. Remarkably, we further show that the distance between such minima and the original training set can be unbounded.

To this end, we present a constructive method for generating new KKT sets from a given one. This method relies on two key lemmas that underpin the construction, after which we show how they can be used to explicitly generate a broad family of KKT sets.

The proofs of lemmas and theorems in this subsection are relegated to Appendix A.

**Lemma 2** (Merge). *Let $S$ be a KKT set and let $\mathbf{x}_1, \mathbf{x}_2 \in S$ be two points with identical labels and activation patterns, and with coefficients $\lambda_1, \lambda_2 > 0$. Then, there exists $\alpha \in (0, 1)$ such that the set $S' := (S \setminus \{\mathbf{x}_1, \mathbf{x}_2\}) \cup \{\mathbf{x}_{1.5}\}$ is also a KKT set, where $\mathbf{x}_{1.5} = \alpha\mathbf{x}_1 + (1 - \alpha)\mathbf{x}_2$.*

**Lemma 3** (Split). *Let $S$ be a KKT set and let $\mathbf{x}_1 \in S$ be a point with coefficient $\lambda_1 > 0$. Then, for all $\alpha, \beta > 0$ and for all $\boldsymbol{\nu} \in \mathbb{R}^d$ such that $\mathbf{z}_1 = \mathbf{x}_1 + \alpha\boldsymbol{\nu}$ and $\mathbf{z}_2 = \mathbf{x}_1 - \beta\boldsymbol{\nu}$ have the same activation pattern and classification as $\mathbf{x}_1$, the set $S' := (S \setminus \{\mathbf{x}_1\}) \cup \{\mathbf{z}_1, \mathbf{z}_2\}$ is a KKT set.*

---

[6]Many of our results can still hold, or be extended, with only partial access to the network's weights. For clarity and simplicity, however, we assume full access in this work.

The above lemmas offer a constructive approach for discovering new global minima of the objective in Equation 6. While the former allows merging two points to create a new KKT set, the latter enables splitting a single point into two. A more visual illustration of this concept can be found in Figure 1.

**Remark 2.** *We stress that the attack in Haim et al. (2022) also applies to certain non-homogeneous networks. Nevertheless, the primary focus of the attack and analysis is on homogeneous neural networks, as they are biased toward meeting these conditions during training.*

While these technical results provide means to explore the loss surface of the objective in Equation 6, they do not provide a quantifiable assessment of the extent to which KKT sets can be manipulated by merging and splitting. While merging imposes few constraints, it provides more limited alterations to the dataset. In contrast, splitting allows flexible alterations but further requires that the new points inherit the activation pattern and classification of the original point from which they are derived, which may constrain their distance to it. Conveniently, under the assumption that the set of training examples does not span the entire data domain, we can show that this distance is unbounded.

**Theorem 4.** *Let $S = \{\mathbf{x}_1, \ldots, \mathbf{x}_n\} \subset \mathbb{R}^d$ be the training set, and let $\Phi(\boldsymbol{\theta}; \mathbf{x})$ a 2-layer neural network that was trained on $S$ and reached a KKT point. If $\mathrm{span}\{\mathbf{x}_1, \ldots, \mathbf{x}_n\} \subsetneq \mathbb{R}^d$, then for all $r > 0$, there exists a KKT set $S_r$ such that $d(S, S_r) > r$. Moreover, all points in $S_r$ are on the margin. That is, $|\Phi(\boldsymbol{\theta}; \mathbf{x}_r)| = p$ for all $\mathbf{x}_r \in S_r$.*

Interestingly, the above theorem implies that the KKT sets that we construct provide a twofold defense: not only are they global minima of the KKT loss, but all points in these sets also lie on the margin. Consequently, even if the margin value $p$ is leaked, the attacker cannot use it to distinguish the actual training set from an arbitrary KKT set.

We stress that, due to Equations 1 and 4, the above theorem pertains only to points on the margin. Thus, as with all other results in this subsection, it continues to hold even if arbitrarily many non-margin points are added to $S$. However, for simplicity, we assume that all points in $S$ lie on the margin.

The assumption that the training examples do not span the entire data domain can be justified by certain real-world datasets that concentrate on low-dimensional structures or even unions of subspaces within the domain, a property commonly exploited in practice (Elhamifar & Vidal, 2013). For example, this is evident in the MNIST dataset, where most images consist primarily of black pixels. Still, in many cases, the data manifold is not a proper subspace of the domain, but instead can be closely approximated by such a subspace. To provide meaningful guarantees in such cases, we present the following theorem.

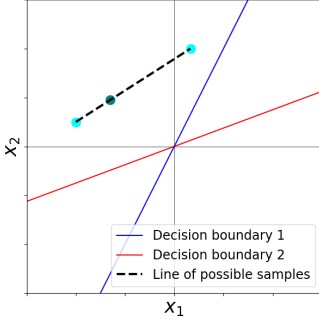

(a) Example of Lemma 2. The green point on the dotted line can replace the two points at its edges.

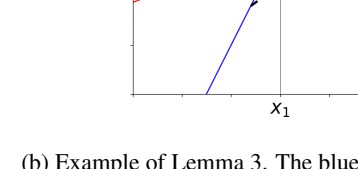

(b) Example of Lemma 3. The blue point can be split into two points along the dotted line.

Figure 1: Illustration of Lemmas 2 and 3 which demonstrate how splitting and merging can allow us to constructively explore the solution space of the attacker's optimization problem, presented in Definition 1. An instance can be split into two instances along a valid direction (Subfigure 1a) or, conversely, two instances can be combined into a single representative instance (Subfigure 1b). Note that the instances obtained through splitting or merging preserve the same activation pattern as the original instance(s).

**Theorem 5.** *Let $S = \{\mathbf{x}_1, \ldots, \mathbf{x}_n\} \subset \mathbb{R}^d$ be a KKT set, let $\Phi(\boldsymbol{\theta}; \mathbf{x})$ be a 2-layer, width-$k$ neural network that was trained on $S$ and reached a KKT point, and suppose that $\gamma > 0$ and $\boldsymbol{\nu} \in \mathbb{R}^d$ satisfy $\langle \boldsymbol{\nu}, \mathbf{x}_i \rangle \leq \gamma$ for all $i$. Then, for all $\mathbf{x}_i \in S$ and any $\alpha, \beta$ not exceeding*

$$\min_{j \in [k]} \frac{|D_j(\mathbf{x}_i)| \cdot \|\mathbf{w}_j\|}{\sum_{i=1}^n \lambda_i} \cdot \frac{1}{\gamma},$$

$S' := (S \setminus \{\mathbf{x}_1\}) \cup \{\mathbf{x}_i + \alpha\boldsymbol{\nu}, \mathbf{x}_i - \beta\boldsymbol{\nu}\}$ *is a KKT set.*

In words, the theorem implies that each point $\mathbf{x}_i$ can be split into two points, each at least $\min_{j \in [k]} \frac{|D_j(\mathbf{x}_i)| \cdot \|\mathbf{w}_j\|}{\sum_{i=1}^n \lambda_i} \cdot \frac{1}{\gamma}$ away from $\mathbf{x}_i$ (larger values of $\alpha$ and $\beta$ may also be admissible). Note that $d \leq n$, since otherwise one could find a vector $\boldsymbol{\nu}$ orthogonal to all data points, giving $\gamma = 0$ and reducing the setting to the assumptions of Theorem 4. We stress that smaller values of $\gamma$ allow points to be split farther – a situation that arises, for example, when the data manifold approximately lies on a linear subspace of the domain. Moreover, we prove that $\gamma \leq \sigma_d$, where $\sigma_d$ is the smallest singular value of the data matrix $S$ written as a column matrix $\mathbf{S} = (\mathbf{x}_1, \ldots, \mathbf{x}_n)$ (see Appendix C for further discussion). Geometrically, this means that the minimal splitting distance is controlled by the "thinnest" direction of the data cloud: the smaller the smallest singular value, the more freedom there is to alter points along directions perpendicular to those where the data is nearly flat.

### 3.2 Analysis beyond the idealized setting: approximate KKT points

In the previous subsection, we assumed that the model converged to a KKT point. Since, in practice, a trained network may only approximate such a stationary point rather than precisely attain it, this subsection provides a more natural yet analytically tractable foundation for studying the associated privacy risks. To that end, we will shortly formulate the following relaxation of the KKT condition.

**Definition 3** (($\varepsilon, \delta$)-KKT – paraphrased from Lyu & Li (2020); Ji & Telgarsky (2020))**.**
*$\boldsymbol{\theta}$ satisfies ($\varepsilon, \delta$)-KKT if the following holds:*

*a)* $\|\boldsymbol{\theta} - \sum_{i=1}^n \lambda_i y_i \nabla_{\boldsymbol{\theta}} \Phi(\boldsymbol{\theta}; \mathbf{x}_i)\|_2 \leq \varepsilon.$     *b)* $\forall i,\ y_i \Phi(\boldsymbol{\theta}; \mathbf{x}_i) \geq p > 0.$
*c)* $\lambda_1, \ldots, \lambda_n \geq 0.$                              *d)* $\forall i,\ \lambda_i(y_i \Phi(\boldsymbol{\theta}; \mathbf{x}_i) - p) \leq \delta.$

Following the above definition, the following is a natural generalization of a KKT set.

**Definition 4** (($\varepsilon, \delta$)-KKT sets)**.** *Let $\Phi(\boldsymbol{\theta}; \mathbf{x})$ be a homogeneous neural network, and let $S = \{\mathbf{x}_1, \ldots, \mathbf{x}_n\}$. We say that $S$ is an ($\varepsilon, \delta$)-KKT set if there exist nonnegative $\lambda_1, \ldots, \lambda_n$ such that $\boldsymbol{\theta}$ is ($\varepsilon, \delta$)-KKT.*

We note that optimizing the objective in Condition a) to accuracy $\varepsilon$ produces ($\varepsilon, \delta$)-KKT sets for some $\delta > 0$. For brevity, when $\delta$ is irrelevant, we simply refer to these as $\varepsilon$-KKT sets. This convention aligns with the setting in Haim et al. (2022), who similarly disregard the $\delta$ term, since it is not used in their attack.

The following are extensions of the merging and splitting lemmas from the previous subsection.

**Lemma 6** (Approximate-KKT merge)**.** *Let $S$ be an $\varepsilon$-KKT set and let $\mathbf{x}_1, \mathbf{x}_2 \in S$ be a point with coefficients $\lambda_1, \lambda_2 > 0$. Then, there exists $\alpha \in (0, 1)$ such that the set $S' = (S \setminus \{\mathbf{x}_1, \mathbf{x}_2\}) \cup \{\alpha\mathbf{x}_1 + (1 - \alpha)\mathbf{x}_2\}$ is also an $\varepsilon$-KKT set.*

**Lemma 7** (Approximate-KKT split)**.** *Let $S$ be an $\varepsilon$-KKT set and let $\mathbf{x}_1 \in S$ be a point with coefficient $\lambda_1 > 0$. Consider the two points: $\mathbf{z}_1 = \mathbf{x}_1 + \alpha\boldsymbol{\nu}$ and $\mathbf{z}_2 = \mathbf{x}_1 - \beta\boldsymbol{\nu}$ where $\alpha, \beta > 0$ and $\boldsymbol{\nu} \in \mathbb{R}^d$ such that $\mathbf{z}_1, \mathbf{z}_2$ have the same activation pattern and classification as $\mathbf{x}_1$. Then, the set $S' = (S \setminus \{\mathbf{x}_1\}) \cup \{\mathbf{z}_1, \mathbf{z}_2\}$ is also an $\varepsilon$-KKT set.*

In the following two subsections, we examine the budget for splitting data points in the almost KKT setting. Each subsection assumes full knowledge of $\varepsilon$ but considers a different level of knowledge the attacker has about $\delta$.

The proofs of lemmas and theorems are relegated to Appendix B.

### 3.2.1 The attacker possesses no information about $\delta$

As a starting point, we first assume in this subsection that the attacker has no information about $\delta$. In such a case, Lemma 7 can still be used as long as the new points are in the same activation pattern as

the original point. The following theorem provides a minimal guarantee on the budget required to split a data point in this setting.

**Theorem 8.** *Let $S = \{\mathbf{x}_1, \ldots, \mathbf{x}_n\} \subset \mathbb{R}^d$ be an $\varepsilon$-KKT set for some $\varepsilon > 0$, let $\Phi(\boldsymbol{\theta}; \mathbf{x})$ be a 2-layer, width-k neural network that was trained on $S$ and reached an $(\varepsilon, \delta)$-KKT point, and suppose that $\gamma > 0$ and $\boldsymbol{\nu} \in \mathbb{R}^d$ satisfy $\langle \boldsymbol{\nu}, \mathbf{x}_i \rangle \leq \gamma$ for all $i$. Then, for all $\mathbf{x}_i \in S$ and any $\alpha, \beta$ not exceeding*

$$\min_{j \in [k]} \frac{|D_j(\mathbf{x}_l)| \|\mathbf{w}_j\|}{\varepsilon + \gamma |v_j| \sum_{i=1}^{n} \lambda_i}, \tag{7}$$

*such that $\mathbf{z}_1 := \mathbf{x}_i + \alpha \boldsymbol{\nu}$ and $\mathbf{z}_2 := \mathbf{x}_i - \beta \boldsymbol{\nu}$ have the same classification as $\mathbf{x}_i$, $S' := (S \setminus \{\mathbf{x}_i\}) \cup \{\mathbf{z}_1, \mathbf{z}_2\}$ is an $\varepsilon$-KKT set.*

Note that when $\gamma = 0$, the above equation simplifies to $\min_{j \in [k]} \frac{|D_j(\mathbf{x}_l)| \|\mathbf{w}_j\|}{\varepsilon}$. Furthermore, if $\varepsilon \to 0$, then both $\alpha$ and $\beta$ become unbounded, consistent with the analogous case in Theorem 5.

### 3.2.2 THE ATTACKER HAS AN UPPER BOUND ON $\delta$

In the previous subsection, we assumed, as an initial test case, that the attacker cannot know $\delta$. This assumption, however, is brittle: in practice, some information about $\delta$ may leak, or the attacker may be able to deduce an upper bound on its value from the training dynamics.[7] Because our splitting technique can degrade the value of $\delta$, this upper bound introduces an additional constraint: we cannot split the training set beyond the attacker's bound, which limits the allowable budget for altering training instances. The following theorem establishes a lower bound on the permissible amount of change in this scenario.

**Theorem 9.** *Let $S = \{\mathbf{x}_1, \ldots, \mathbf{x}_n\} \subset \mathbb{R}^d$ be an $(\varepsilon, \delta)$-KKT set for some $\varepsilon, \delta > 0$, and let $\Phi(\boldsymbol{\theta}; \mathbf{x})$ be a 2-layer, width-k neural network that was trained on $S$ and reached an $(\varepsilon, \delta)$-KKT point. Suppose that $\gamma > 0$ and $\boldsymbol{\nu} \in \mathbb{R}^d$ satisfy $\langle \boldsymbol{\nu}, \mathbf{x}_i \rangle \leq \gamma$ for all $i$, and consider the dataset $S' = (S \setminus \{\mathbf{x}_1\}) \cup \{\mathbf{x}_l + \alpha_l \boldsymbol{\nu}, \mathbf{x}_l - \beta_l \boldsymbol{\nu}\}$ for $\alpha$ and $\beta$ not exceeding Equation 7. Then, if $\Delta \delta < p$, $S'$ is an $(\varepsilon, \delta + \Delta \delta)$-KKT set, where*

$$\Delta \delta := \lambda_l (\alpha_l + \beta_l) \sum_{j \in J} |v_j| \left( \varepsilon + \gamma |v_j| \sum_{i=1}^{n} \lambda_i \right).$$

When $\gamma = 0$, the expression simplifies to $\delta + \varepsilon \lambda_l (\alpha_l + \beta_l) \sum_{j=1}^{k} |v_j|$. Thus, although $\delta$ deteriorates with increased splitting, the effectiveness of the reconstruction attack diminishes as $\gamma$ decreases and the network approaches a KKT point, indicating that for well-trained networks on structured data, such attacks become unreliable.

## 4 EXPERIMENTS

In this section, we present experiments that complement our theoretical analysis. Our theory shows that the objective in Equation 6 admits ubiquitous global minima under certain conditions, but this alone does not guarantee that the attack will avoid converging to the true training set. To examine this, we empirically test whether training-set leakage can occur without prior knowledge. We model the attacker's prior as knowledge of the data domain boundaries and incorporate this into the attack through the initialization distribution. For instance, when the domain consists of natural images, the prior encodes that the data lie within the valid pixel range $[0, 1]^d$, so the attacker initializes only within this range. This type of prior was also used by Haim et al. (2022), who employed both a correctly scaled initialization and an optimization penalty to ensure solutions stayed within the natural-image domain.

**No leakage without prior.** We generated 500 synthetic training samples (250 per class) uniformly on the unit sphere $\mathbb{S}^{783} \subset \mathbb{R}^{784}$, and labeled them according to the sign of the first coordinate. We then trained a 2-layer, width-1,000 ReLU network on this data for 500K epochs, achieving a final

---

[7]For instance, a large value of $\delta$ may indicate that the training process terminated prematurely, leading to poor performance.

training loss of $10^{-7}$. To assess reconstructability, we initialize the candidate reconstructions $\mathbf{x}_i$ on spheres of varying radii centered at the origin (each corresponding to a different level of prior information available to the attacker). For each setting, we record the average distance between the 5 best reconstructions and the actual training set over multiple runs.

We evaluate two types of prior. The first one is the magnitude of the data, and the second one is its geometry. Spheres with different radii have different geometries, such as different curvatures, different typical inner products, and different typical distances between two randomly sampled points.

Although all runs achieve similar KKT objective values (ranging from 330 to 332), they yield markedly different reconstruction qualities, revealing a strong dependence on initialization and convergence to distinct minima. In the absence of prior information, near-optimal solutions thus do not reliably recover the original samples. Figure 2a depicts the distribution of the average distance between the training set and the best five reconstruction attempts. It demonstrates that the reconstruction error increases as the assumed radius deviates from the true domain of the data. Consequently, successful reconstruction appears to strongly depend on prior knowledge of the data domain.

**Beyond the theoretical framework.** We trained the same 3-layer architecture on CIFAR as was done by Haim et al. (2022), after shifting all training samples by various magnitudes, corresponding to different levels of prior information available to the attacker.[8] To reconstruct the training set, we minimized the objective in Equation 6, without incorporating any additional regularization.

As shown in Figure 2b, the results quickly deteriorate as the attacker's prior weakens. Moreover, the reconstructions clearly resemble averages of multiple training instances, indicating that the minimum reached by the attack's optimization procedure is in fact an interpolation of several training examples, as predicted by our theory.

Prior knowledge is essential for successful reconstruction attacks; KKT conditions alone do not necessarily reveal information about the training data. Experiments show that attack effectiveness is directly linked to the strength of prior knowledge: stronger/more accurate priors lead to stronger attacks, making an adaptive adversary that can estimate the true prior highly effective. Privacy risks are largely mitigated if the training data is shifted by a secret bias, even when the general data domain is known. Estimating and defending against such adaptive priors is an interesting direction for future work.

For more experiments on various datasets and architectures, we refer the reader to Appendix F.

## 5    SUMMARY AND DISCUSSION

In this paper, we demonstrated both theoretically and empirically that the objective function underlying the reconstruction attack of Haim et al. (2022) admits ubiquitous global minima. Consequently, reconstruction is generally unreliable without prior knowledge, which suggests new avenues for mitigating such attacks, for example, by shifting the training set with a secret bias. Notably, our results indicate that the implicit bias induced by gradient methods can actually prevent leakage rather than facilitate it, which may seem counterintuitive in light of previous work.

Although our proposed defenses are theoretically motivated, they do not provably preclude reconstruction, since an attacker might still infer information about the data domain directly from the model. We leave the intriguing question of whether this is indeed possible and how to design provably secure defenses for future work. Expanding our findings to more modern, non-homogeneous neural networks is an important and interesting question for future work.

---

[8]While conceptually equivalent, this is not precisely how we obtained our network. See Appendix D for further details.

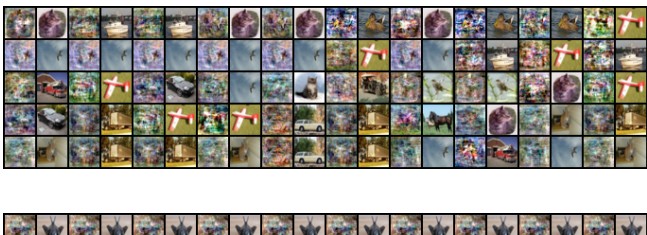

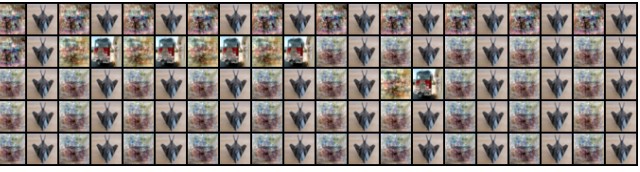

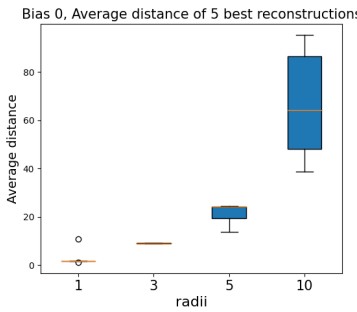

(a) A comparison of the reconstruction results based on initializations with different radii, for a network that was trained with data sampled from the unit sphere, and measured by the average Euclidean distance of the 5 best reconstructions. The radius increases as the prior knowledge available to the attacker weakens, significantly degrading the quality of the reconstruction.

(b) Reconstructed CIFAR images (odd columns) and their training set nearest neighbors (even columns). We trained a non-regularized model on data shifted by 0.5 (top image) and 5 (bottom image) until reaching an almost-KKT point. The experiment demonstrates that attack effectiveness rapidly diminishes with increasing data shift, indicating a weaker prior for the attacker. While the top reconstruction captures a small subset of the training data, the bottom reconstruction fails entirely.

Figure 2: The left figure shows the attack on the unit sphere. The images on the right show the attack on CIFAR with shifted training data.

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

## A  PROOFS FOR SUBSECTION 3.1

In this section, we present all the proofs for Subsection 3.1. We begin with a few additional notations that will be used throughout the appendix.

Given a neuron with weights $\mathbf{w}$ and bias $b$, define the shorthand $c_j(\mathbf{x}) \coloneqq [\langle \mathbf{w}_j, \mathbf{x} \rangle + b_j]_+$. Let $\sigma\prime_j$ denote a subgradient of $\left[ \mathbf{w}_j^\top \mathbf{x} + b_j \right]_+$; for sample $\mathbf{x}_i$, write $\sigma'_{i,j}$ for the subgradient of $\left[ \mathbf{w}_j^\top \mathbf{x}_i + b_j \right]_+$.

### A.1  PROOFS OF LEMMA 2 AND LEMMA 3

We begin with proving an auxiliary lemma about the convex nature of the gradient of the neural network, and then we prove Lemmas 2 and 3.

**Lemma 10.** *Let $\Phi(\boldsymbol{\theta}; \mathbf{x})$ be a neural network with piecewise-linear activations. Suppose that the point $\alpha \mathbf{x}_1 + (1 - \alpha)\mathbf{x}_2$ is at the same activation pattern as $\mathbf{x}_1$ and $\mathbf{x}_2$. Then, $\nabla_{\boldsymbol{\theta}} \Phi(\boldsymbol{\theta}; \mathbf{x})$ is convex with respect to $\mathbf{x}$:*

$$\nabla_{\boldsymbol{\theta}} \Phi(\boldsymbol{\theta}; \alpha \mathbf{x}_1 + (1 - \alpha)\mathbf{x}_2) = \alpha \nabla_{\boldsymbol{\theta}} \Phi(\boldsymbol{\theta}; \mathbf{x}_1) + (1 - \alpha)\nabla_{\boldsymbol{\theta}} \Phi(\boldsymbol{\theta}; \mathbf{x}_2).$$

*Proof.* Let $\Phi(\boldsymbol{\theta}; \mathbf{x})$ be a neural network with ReLU activations (the argument is the same for other piecewise linear functions), then since the gradient $\nabla_{\boldsymbol{\theta}} \Phi(\boldsymbol{\theta}; \mathbf{x})$ is with respect to $\boldsymbol{\theta}$, it remains an affine function in $\mathbf{x}$ at each activation pattern, and can thus be written as $\nabla_{\boldsymbol{\theta}} \Phi(\boldsymbol{\theta}; \mathbf{x}) = \mathbf{A}\mathbf{x} + \mathbf{c}$ for some matrix $\mathbf{A}$ and vector $\mathbf{c}$. We therefore have

$$\begin{aligned}
\alpha \nabla_{\boldsymbol{\theta}} \Phi(\boldsymbol{\theta}; \mathbf{x}_1) + (1 - \alpha)\nabla_{\boldsymbol{\theta}} \Phi(\boldsymbol{\theta}; \mathbf{x}_2) &= \alpha(\mathbf{A}\mathbf{x}_1 + \mathbf{c}) + (1 - \alpha)(\mathbf{A}\mathbf{x}_2 + \mathbf{c}) \\
&= \mathbf{A}(\alpha \mathbf{x}_1 + (1 - \alpha)\mathbf{x}_2) + \mathbf{c} \\
&= \nabla_{\boldsymbol{\theta}} \Phi(\boldsymbol{\theta}; \alpha \mathbf{x}_1 + (1 - \alpha)\mathbf{x}_2).
\end{aligned}$$

$\square$

We now turn to prove the lemmas.

*Proof of Lemma 2.* Denote $\lambda' = \lambda_1 + \lambda_2$ and $\alpha = \frac{\lambda_1}{\lambda_1 + \lambda_2}$. It is easy to see that $\alpha \in (0, 1)$ and that $\lambda' > 0$, which proves that Equations 3 and 4 hold. Moreover, since $\mathbf{x}_1$ and $\mathbf{x}_2$ have the same activation

pattern, we have that $\mathbf{x}_{1.5}$ also has the same activation pattern since it is a convex combination of the two, which proves that Equation 2 holds. To show that Equation 1 holds, we compute

$$
\begin{aligned}
\boldsymbol{\theta} &= \sum_{i=1}^{n} \lambda_i y_i \nabla_{\boldsymbol{\theta}} \Phi(\boldsymbol{\theta}; \mathbf{x}_i) \\
&= \sum_{i \neq 1,2} \lambda_i y_i \nabla_{\boldsymbol{\theta}} \Phi(\boldsymbol{\theta}; \mathbf{x}_i) + \lambda_1 y_1 \nabla_{\boldsymbol{\theta}} \Phi(\boldsymbol{\theta}; \mathbf{x}_1) + \lambda_2 y_2 \nabla_{\boldsymbol{\theta}} \Phi(\boldsymbol{\theta}; \mathbf{x}_2) \\
&\overset{*}{=} \sum_{i \neq 1,2} \lambda_i y_i \nabla_{\boldsymbol{\theta}} \Phi(\boldsymbol{\theta}; \mathbf{x}_i) + \lambda_1 y_{1.5} \nabla_{\boldsymbol{\theta}} \Phi(\boldsymbol{\theta}; \mathbf{x}_1) + \lambda_2 y_{1.5} \nabla_{\boldsymbol{\theta}} \Phi(\boldsymbol{\theta}; \mathbf{x}_2) \\
&\overset{**}{=} \sum_{i \neq 1,2} \lambda_i y_i \nabla_{\boldsymbol{\theta}} \Phi(\boldsymbol{\theta}; \mathbf{x}_i) + y_{1.5} \nabla_{\boldsymbol{\theta}} \Phi(\boldsymbol{\theta}; \lambda_1 \mathbf{x}_1 + \lambda_2 \mathbf{x}_2) \\
&= \sum_{i \neq 1,2} \lambda_i y_i \nabla_{\boldsymbol{\theta}} \Phi(\boldsymbol{\theta}; \mathbf{x}_i) + y_{1.5} \nabla_{\boldsymbol{\theta}} \Phi\left(\boldsymbol{\theta}; (\lambda_1 + \lambda_2)\left(\frac{\lambda_1}{\lambda_1 + \lambda_2}\mathbf{x}_1 + \frac{\lambda_2}{\lambda_1 + \lambda_2}\mathbf{x}_2\right)\right) \\
&= \sum_{i \neq 1,2} \lambda_i y_i \nabla_{\boldsymbol{\theta}} \Phi(\boldsymbol{\theta}; \mathbf{x}_i) + y_{1.5} \nabla_{\boldsymbol{\theta}} \Phi(\boldsymbol{\theta}; \lambda'(\alpha \mathbf{x}_1 + (1-\alpha)\mathbf{x}_2)) \\
&= \sum_{i \neq 1,2} \lambda_i y_i \nabla_{\boldsymbol{\theta}} \Phi(\boldsymbol{\theta}; \mathbf{x}_i) + y_{1.5} \lambda' \nabla_{\boldsymbol{\theta}} \Phi(\boldsymbol{\theta}; \mathbf{x}_{1.5}),
\end{aligned}
$$

where $*$ is by the assumption that $y_1 = y_2 = y_{1.5}$, and $**$ is by Lemma 10. $\square$

*Proof of Lemma 3.* Denote $\lambda_\alpha = \frac{\alpha \lambda_1}{\alpha + \beta}$, $\lambda_\beta = \frac{\beta \lambda_1}{\alpha + \beta}$. It is easy to see that $\lambda_\alpha, \lambda_\beta > 0$, which proves that Equations 3 and 4 hold. Moreover, since by assumption $\mathbf{z}_1$ and $\mathbf{z}_2$ are split such that they have the same activation pattern as $\mathbf{x}_1$, this assures that Equation 2 holds. Now, observe that

1. $\lambda_\beta + \lambda_\alpha = \frac{\lambda_1 \beta}{\alpha + \beta} + \frac{\lambda_1 \alpha}{\alpha + \beta} = \lambda_1$,

2. $\lambda_\beta(\mathbf{x}_1 + \alpha \boldsymbol{\nu}) + \lambda_\alpha(\mathbf{x}_1 - \beta \boldsymbol{\nu}) = (\lambda_\beta + \lambda_\alpha)\mathbf{x}_1 + \frac{\lambda_1 \beta}{\alpha + \beta}\alpha \boldsymbol{\nu} - \frac{\lambda_1 \alpha}{\alpha + \beta}\beta \boldsymbol{\nu} = \lambda_1 \mathbf{x}_1$.

Then, by the above, we have

$$
\begin{aligned}
\lambda_\beta \nabla_{\boldsymbol{\theta}} \Phi(\boldsymbol{\theta}; \mathbf{z}_1) + \lambda_\alpha \nabla_{\boldsymbol{\theta}} \Phi(\boldsymbol{\theta}; \mathbf{z}_2) &= \lambda_\beta(\mathbf{A}(\mathbf{x}_1 + \alpha \boldsymbol{\nu}) + \mathbf{c}) + \lambda_\alpha(\mathbf{A}(\mathbf{x}_1 - \beta \boldsymbol{\nu}) + \mathbf{c}) \\
&= (\lambda_\beta + \lambda_\alpha)\mathbf{A}\mathbf{x}_1 + (\lambda_\alpha + \lambda_\beta)\mathbf{c} \\
&= \lambda_1(\mathbf{A}\mathbf{x}_1 + \mathbf{c}) \\
&= \lambda_1 \nabla_{\boldsymbol{\theta}} \Phi(\boldsymbol{\theta}; \mathbf{x}_1). \quad (8)
\end{aligned}
$$

Next, let us see that indeed Equation 1 holds by computing

$$
\begin{aligned}
\boldsymbol{\theta} &= \sum_{i=1}^{n} \lambda_i y_i \nabla_{\boldsymbol{\theta}} \Phi(\boldsymbol{\theta}; \mathbf{x}_i) \\
&= \sum_{i \neq 1} \lambda_i y_i \nabla_{\boldsymbol{\theta}} \Phi(\boldsymbol{\theta}; \mathbf{x}_i) + \lambda_1 y_1 \nabla_{\boldsymbol{\theta}} \Phi(\boldsymbol{\theta}; \mathbf{x}_1) \\
&\overset{*}{=} \sum_{i \neq 1} \lambda_i y_i \nabla_{\boldsymbol{\theta}} \Phi(\boldsymbol{\theta}; \mathbf{x}_i) + y_1 \lambda_\beta \nabla_{\boldsymbol{\theta}} \Phi(\boldsymbol{\theta}; \mathbf{z}_1) + y_1 \lambda_\alpha \nabla_{\boldsymbol{\theta}} \Phi(\boldsymbol{\theta}; \mathbf{z}_2),
\end{aligned}
$$

where $*$ is by Equation 8. Thus, the set $S'$ is a KKT set. $\square$

## A.2 PROOFS OF THEOREM 4 AND THEOREM 5

We begin with proving the following auxiliary lemmas. Lemma 11 provides applicable technical constraints that any KKT network must satisfy, and Lemma 12 shows that we can split the training samples in a single direction, and that all points in the new KKT set will remain on the margin. Lemma 13 assures that we can split along this direction as far as we want and remain in the same activation pattern as the original sample.

**Lemma 11.** *Suppose a 2-layer, homogeneous ReLU neural network with parameters $\boldsymbol{\theta}$ that satisfies the KKT conditions in Equations 1-4. Then we have*

$$v_j = \sum_{i=1}^{n} \lambda_i y_i \left[ \mathbf{w}_j^\top \mathbf{x}_i + b_j \right]_+, \quad \mathbf{w}_j = v_j \sum_{i=1}^{n} \lambda_i y_i \mathbf{x}_i \sigma'_{i,j}, \quad b_j = v_j \sum_{i=1}^{n} \lambda_i y_i \sigma'_{i,j}.$$

*Proof.* We compute the derivatives with respect to $\boldsymbol{\theta}$ as follows

$$\frac{\partial}{\partial v_j} \Phi(\boldsymbol{\theta}; \mathbf{x}) = \left[ \mathbf{w}_j^\top \mathbf{x} + b_j \right]_+, \quad \frac{\partial}{\partial \mathbf{w}_j} \Phi(\boldsymbol{\theta}; \mathbf{x}) = v_j x \sigma'_j, \quad \frac{\partial}{\partial b_j} \Phi(\boldsymbol{\theta}; \mathbf{x}) = v_j \sigma'_j.$$

If $\mathbf{w}_j^\top \mathbf{x} + b_j \neq 0$ then $\sigma'_j$ is well defined, and if $\mathbf{w}_j^\top \mathbf{x} + b_j = 0$ then $\sigma'_j \in [0,1]$. In any case, it holds that $\sigma'_j \geq 0$. Combining the above partial derivatives with Equation 1, we obtain

$$v_j = \sum_{i=1}^{n} \lambda_i y_i \left[ \mathbf{w}_j^\top \mathbf{x}_i + b_j \right]_+, \quad \mathbf{w}_j = v_j \sum_{i=1}^{n} \lambda_i y_i \mathbf{x}_i \sigma'_{i,j}, \quad b_j = v_j \sum_{i=1}^{n} \lambda_i y_i \sigma'_{i,j}$$

for all $j \in [k]$, as required. $\qquad\square$

**Lemma 12.** *Let $S = \{\mathbf{x}_1, \ldots, \mathbf{x}_n\}$ be a KKT set such that all $\mathbf{x}_i \in S$ are on the margin, let $p$ be the margin's value, and let $\hat{\mathbf{x}}$ be a vector that is orthogonal to all $\mathbf{x}_i \in S$. Then, for all $\beta \in \mathbb{R}$ and for all $l \in [n]$ we have that $|\Phi(\boldsymbol{\theta}; \mathbf{x}_l + \beta\hat{\mathbf{x}})| = p$.*

*Proof.* Compute

$$|\Phi(\boldsymbol{\theta}; \mathbf{x}_l + \beta\hat{\mathbf{x}})| = \left| \sum_{j=1}^{k} v_j \left[ \langle \mathbf{w}_j, \mathbf{x}_l + \beta\hat{\mathbf{x}} \rangle + b_j \right]_+ \right| \overset{*}{=} \left| \sum_{j=1}^{k} v_j \left[ v_j \sum_{i=1}^{n} \lambda_i y_i \sigma'_{i,j} \langle \mathbf{x}_i, \mathbf{x}_l + \beta\hat{\mathbf{x}} \rangle + b_j \right]_+ \right|$$

$$\overset{**}{=} \left| \sum_{j=1}^{k} v_j \left[ v_j \sum_{i=1}^{n} \lambda_i y_i \sigma'_{i,j} \langle \mathbf{x}_i, \mathbf{x}_l \rangle + b_j \right]_+ \right| = \left| \sum_{j=1}^{k} v_j \left[ \left\langle v_j \sum_{i=1}^{n} \lambda_i y_i \sigma'_{i,j} \mathbf{x}_i, \mathbf{x}_l \right\rangle + b_j \right]_+ \right|$$

$$= \left| \sum_{j=1}^{k} v_j \left[ \langle \mathbf{w}_j, \mathbf{x}_l \rangle + b_j \right]_+ \right| = |\Phi(\boldsymbol{\theta}; \mathbf{x}_l)| = p.$$

Where $*$ is due to Lemma 11, and $**$ is due to the fact that $\hat{\mathbf{x}}$ is orthogonal to all $\mathbf{x}_i$. $\qquad\square$

**Lemma 13.** *Let $S = \{\mathbf{x}_1, \ldots, \mathbf{x}_n\}$ be a KKT set and let $\hat{\mathbf{x}}$ be a vector that is orthogonal to all $\mathbf{x}_i \in S$. Then, $c_j(\mathbf{x}_l + \beta\hat{\mathbf{x}}) = c_j(\mathbf{x}_l)$ for all $l \in [n]$ and for all $\beta \in \mathbb{R}$.*

*Proof.* Compute

$$c_j(\mathbf{x}_l + \beta\hat{\mathbf{x}}) = \left[ \langle \mathbf{w}_j, \mathbf{x}_l + \beta\hat{\mathbf{x}} \rangle + b_j \right]_+ \overset{*}{=} \left[ v_j \sum_{i=1}^{n} \lambda_i y_i \sigma'_{i,j} \langle \mathbf{x}_i, \mathbf{x}_l + \beta\hat{\mathbf{x}} \rangle + b_j \right]_+$$

$$\overset{**}{=} \left[ v_j \sum_{i=1}^{n} \lambda_i y_i \sigma'_{i,j} \langle \mathbf{x}_i, \mathbf{x}_l \rangle + b_j \right]_+ \overset{***}{=} \left[ \langle \mathbf{w}_j, \mathbf{x}_l \rangle + b_j \right]_+ = c_j(\mathbf{x}_l),$$

where $*$ and $***$ use Lemma 11, and $**$ uses that fact that $\hat{\mathbf{x}}$ is orthogonal to all $\mathbf{x}_i$. $\qquad\square$

Using these two lemmas, we can now prove Theorem 4.

*Proof of Theorem 4.* Let $S = \{\mathbf{x}_1, \ldots, \mathbf{x}_n\}$ be a KKT set. Since $\text{span}\{\mathbf{x}_1, \ldots, \mathbf{x}_n\} \subsetneq \mathbb{R}^d$, there exists a vector $\hat{\mathbf{x}} \in \mathbb{R}^d$ that is orthogonal to all $\mathbf{x}_l \in S$. Since $\hat{\mathbf{x}}$ is a directional vector, we can assume without loss of generality that $\|\hat{\mathbf{x}}\| = 1$. For each $\mathbf{x}_l \in S$ we define two new points $\mathbf{x}_{l_1} = \mathbf{x}_l + \alpha_l \hat{\mathbf{x}}$ and $\mathbf{x}_{l_2} = \mathbf{x}_l - \beta_l \hat{\mathbf{x}}$ for some $\alpha_l, \beta_l > 0$. We need to show that the set $S' = \bigcup_{l=1}^{n} \{\mathbf{x}_{l_1}, \mathbf{x}_{l_2}\}$ is a KKT set. Using Lemma 3 (which we can use since we know that $\mathbf{x}_l$ and $\{\mathbf{x}_{l_1}, \mathbf{x}_{l_2}\}$ have the same

activation pattern from Lemma 13, and also the same classification by assumption), we can replace each instance $\mathbf{x}_l \in S$ with $\mathbf{x}_{l_1}, \mathbf{x}_{l_2}$ iteratively until we get $S'$. This proves that $S'$ is a KKT set. Moreover, by Lemma 12 we have that all points in $S'$ are on the margin. Lastly, we need to prove that for any distance $\tau > 0$ we can chose $\alpha_l, \beta_l$ such that $d(S, S') > \tau$. Let us compute the distance of some $\mathbf{x}_{l_1} = \mathbf{x}_l + \alpha_l \hat{\mathbf{x}}$ from all points $\mathbf{x}_i \in S$ as follows

$$
\begin{aligned}
\|\mathbf{x}_{l_1} - \mathbf{x}_i\|_2^2 &= \|\mathbf{x}_{l_1}\|^2 - 2\langle \mathbf{x}_{l_1}, \mathbf{x}_i \rangle + \|\mathbf{x}_i\|^2 \\
&= \|\mathbf{x}_l + \alpha_l \hat{\mathbf{x}}\|^2 - 2\langle \mathbf{x}_l + \alpha_l \hat{\mathbf{x}}, \mathbf{x}_i \rangle + \|\mathbf{x}_i\|^2 \\
&= \|\mathbf{x}_l + \alpha_l \hat{\mathbf{x}}\|^2 - 2\langle \mathbf{x}_l, \mathbf{x}_i \rangle + \|\mathbf{x}_i\|^2 \\
&= \|\mathbf{x}_l\|^2 + 2\alpha_l \langle \mathbf{x}_l, \hat{\mathbf{x}} \rangle + \alpha_l^2 \|\hat{\mathbf{x}}\|^2 - 2\langle \mathbf{x}_l, \mathbf{x}_i \rangle + \|\mathbf{x}_i\|^2 \\
&= \|\mathbf{x}_l\|^2 + \alpha_l^2 \|\hat{\mathbf{x}}\|^2 - 2\langle \mathbf{x}_l, \mathbf{x}_i \rangle + \|\mathbf{x}_i\|^2 \\
&= \|\mathbf{x}_l\|^2 + \alpha_l^2 - 2\langle \mathbf{x}_l, \mathbf{x}_i \rangle + \|\mathbf{x}_i\|^2.
\end{aligned}
$$

Since $\mathbf{x}_l$ and $\mathbf{x}_i$ are constants, then $\|\mathbf{x}_l\|^2$, $2\langle \mathbf{x}_l, \mathbf{x}_i \rangle$ and $\|\mathbf{x}_i\|^2$ are also constants. The only parameter we can change is $\alpha_l$, so we can take $\alpha_l$ large enough to make sure that $d(\mathbf{x}_{l_1}, \mathbf{x}_i) > \tau$ for all $\mathbf{x}_i \in S$. The same analysis also shows that this approach can be applied iteratively to all remaining instances. Note that we can take $\alpha_l$ and $\beta_l$ to be as large as we want since the new points $\mathbf{x}_{l_1}$ and $\mathbf{x}_{l_2}$ will still have the same activation pattern as $\mathbf{x}_l$ (for all $\mathbf{x}_l \in S$) by virtue of Lemma 13. We can thus bound the distance between $S$ and $S'$) by

$$
d(S, S') = \min_{\mathbf{x} \in S, \mathbf{x}' \in S'} \|\mathbf{x} - \mathbf{x}'\| > \tau.
$$

$\square$

We now turn to proving Theorem 5.

*Proof of Theorem 5.* Let $\mathbf{x}_l \in S$ and let $\mathbf{x}_l + \alpha\boldsymbol{\nu}$ and $\mathbf{x}_l - \beta\boldsymbol{\nu}$ be the new points resulted from splitting $\mathbf{x}_l$. Both points have to remain in the same activation pattern as $\mathbf{x}_l$, meaning that for each neuron $c_j(\mathbf{x})$, we need to make sure that $\text{sign}(c_j(\mathbf{x}_l)) = \text{sign}(c_j(\mathbf{x}_l + \alpha\boldsymbol{\nu})) = \text{sign}(c_j(\mathbf{x}_l - \beta\boldsymbol{\nu}))$. Namely, that we do not change the ReLU activation of any neuron $\mathbf{w}_j^\top \mathbf{x} + b_j$. $\mathbf{x}_l + \alpha\boldsymbol{\nu}$ changes an activation when $\langle \mathbf{w}_j, \mathbf{x}_l + \alpha\boldsymbol{\nu} \rangle + b_j = 0$, implying that

$$
\begin{aligned}
\langle \mathbf{w}_j, \mathbf{x}_l + \alpha\boldsymbol{\nu} \rangle + b_j = 0 &\Rightarrow \langle \mathbf{w}_j, \mathbf{x}_l \rangle + b_j + \alpha \langle \mathbf{w}_j, \boldsymbol{\nu} \rangle = 0 \\
&\Rightarrow \alpha = -\frac{\langle \mathbf{w}_j, \mathbf{x}_l \rangle + b_j}{\langle \mathbf{w}_j, \boldsymbol{\nu} \rangle}.
\end{aligned}
$$

Now we can upper bound the magnitude of $\alpha$ to see how far away $\mathbf{x}_l + \alpha\boldsymbol{\nu}$ can be from $\mathbf{x}_l$. Let us denote the signed distance between a point $\mathbf{x}$ and the hyperplane $\langle \mathbf{w}_j, \mathbf{x} \rangle + b_j = 0$ by $D_j(\mathbf{x}) = \frac{\langle \mathbf{w}_j, \mathbf{x} \rangle + b_j}{\|\mathbf{w}_j\|}$. We can rewrite $\alpha = -\frac{D_j(\mathbf{x}_l)\|\mathbf{w}_j\|}{\langle \mathbf{w}_j, \boldsymbol{\nu} \rangle}$. Now let us bound $|\langle \mathbf{w}_j, \boldsymbol{\nu} \rangle|$ as follows

$$
\begin{aligned}
|\langle \mathbf{w}_j, \boldsymbol{\nu} \rangle| &= \left| \left\langle \sum_{i=1}^n \lambda_i y_i \sigma'_{i,j} \mathbf{x}_i, \boldsymbol{\nu} \right\rangle \right| \\
&= \left| \sum_{i=1}^n \lambda_i y_i \sigma'_{i,j} \langle \mathbf{x}_i, \boldsymbol{\nu} \rangle \right| \\
&\leq \sum_{i=1}^n \lambda_i |y_i| |\sigma'_{i,j}| |\langle \mathbf{x}_i, \boldsymbol{\nu} \rangle| \\
&\leq \gamma \sum_{i=1}^n \lambda_i.
\end{aligned}
$$

The above implies that

$$
\alpha \geq \frac{|D_j(\mathbf{x}_l)| \cdot \|\mathbf{w}_j\|}{\gamma \sum_{i=1}^n \lambda_i} = \frac{|D_j(\mathbf{x}_l)| \cdot \|\mathbf{w}_j\|}{\sum_{i=1}^n \lambda_i} \cdot \frac{1}{\gamma}.
$$

This is a sufficient condition on all neurons $c_j$ to guarantee that we do not deviate from our current activation pattern. This means that any $\alpha$ not exceeding

$$\min_{j \in [k]} \frac{|D_j(\mathbf{x}_l)| \cdot \|\mathbf{w}_j\|}{\sum_{i=1}^n \lambda_i} \cdot \frac{1}{\gamma}$$

is a valid choice, where the exact same analysis gives the same bound for $\beta$. $\qquad\square$

## B  PROOFS FOR SUBSECTION 3.2

### B.1  PROOFS OF LEMMA 6 AND LEMMA 7

*Proof of Lemma 6.* Define $\lambda' := \lambda_1 + \lambda_2$ and $\alpha := \frac{\lambda_1}{\lambda_1 + \lambda_2}$, $\mathbf{x}_{1.5} := \alpha \mathbf{x}_1 + (1 - \alpha)\mathbf{x}_2$. It is easy to see that $\alpha \in (0, 1)$ and $\lambda' > 0$, which proves that Condition c) holds. Moreover, Condition d) is also easily satisfied for a sufficiently large $\delta$, and Condition b) holds since the merged point $\mathbf{x}_{1.5}$ is a convex combination of $\mathbf{x}_1, \mathbf{x}_2$ and thus has the same activation pattern and classification as them. To show Condition a), we compute

$$
\begin{aligned}
\sum_{i=1}^n \lambda_i y_i \nabla_{\boldsymbol{\theta}} \Phi(\boldsymbol{\theta}; \mathbf{x}_i) &= \sum_{i \neq 1,2} \lambda_i y_i \nabla_{\boldsymbol{\theta}} \Phi(\boldsymbol{\theta}; \mathbf{x}_i) + \lambda_1 y_1 \nabla_{\boldsymbol{\theta}} \Phi(\boldsymbol{\theta}; \mathbf{x}_1) + \lambda_2 y_2 \nabla_{\boldsymbol{\theta}} \Phi(\boldsymbol{\theta}; \mathbf{x}_2) \\
&\overset{*}{=} \sum_{i \neq 1,2} \lambda_i y_i \nabla_{\boldsymbol{\theta}} \Phi(\boldsymbol{\theta}; \mathbf{x}_i) + \lambda_1 y_{1.5} \nabla_{\boldsymbol{\theta}} \Phi(\boldsymbol{\theta}; \mathbf{x}_1) + \lambda_2 y_{1.5} \nabla_{\boldsymbol{\theta}} \Phi(\boldsymbol{\theta}; \mathbf{x}_2) \\
&\overset{**}{=} \sum_{i \neq 1,2} \lambda_i y_i \nabla_{\boldsymbol{\theta}} \Phi(\boldsymbol{\theta}; \mathbf{x}_i) + y_{1.5} \nabla_{\boldsymbol{\theta}} \Phi(\boldsymbol{\theta}; \lambda_1 \mathbf{x}_1 + \lambda_2 \mathbf{x}_2) \\
&= \sum_{i \neq 1,2} \lambda_i y_i \nabla_{\boldsymbol{\theta}} \Phi(\boldsymbol{\theta}; \mathbf{x}_i) + y_{1.5} \nabla_{\boldsymbol{\theta}} \Phi\left(\boldsymbol{\theta}; (\lambda_1 + \lambda_2)\left(\frac{\lambda_1}{\lambda_1 + \lambda_2} \mathbf{x}_1 + \frac{\lambda_2}{\lambda_1 + \lambda_2} \mathbf{x}_2\right)\right) \\
&= \sum_{i \neq 1,2} \lambda_i y_i \nabla_{\boldsymbol{\theta}} \Phi(\boldsymbol{\theta}; \mathbf{x}_i) + y_{1.5} \nabla_{\boldsymbol{\theta}} \Phi(\boldsymbol{\theta}; \lambda'(\alpha \mathbf{x}_1 + (1 - \alpha)\mathbf{x}_2)) \\
&= \sum_{i \neq 1,2} \lambda_i y_i \nabla_{\boldsymbol{\theta}} \Phi(\boldsymbol{\theta}; \mathbf{x}_i) + y_{1.5} \lambda' \nabla_{\boldsymbol{\theta}} \Phi(\boldsymbol{\theta}; \mathbf{x}_{1.5}),
\end{aligned}
$$

where $*$ is from the assumption that $y_1 = y_2 = y_{1.5}$, and $**$ follows from Lemma 10, implying that

$$\|\boldsymbol{\theta} - \sum_{i \neq 1,2} \lambda_i y_i \nabla_{\boldsymbol{\theta}} \Phi(\boldsymbol{\theta}; \mathbf{x}_i) - y_{1.5} \lambda' \nabla_{\boldsymbol{\theta}} \Phi(\boldsymbol{\theta}; \mathbf{x}_{1.5})\|^2 = \|\boldsymbol{\theta} - \sum_{i=1}^n \lambda_i y_i \nabla_{\boldsymbol{\theta}} \Phi(\boldsymbol{\theta}; \mathbf{x}_i)\|^2 \leq \varepsilon,$$

and therefore $S'$ is also an $\varepsilon$-KKT set. $\qquad\square$

*Proof of Lemma 7.* Denote $\lambda_\alpha = \frac{\alpha \lambda_1}{\alpha + \beta}$, $\lambda_\beta = \frac{\beta \lambda_1}{\alpha + \beta}$. It is easy to see that $\lambda_\alpha, \lambda_\beta > 0$, which proves that Condition c) holds. Moreover, Condition d) is also easily satisfied for a sufficiently large $\delta$, and Condition b) holds by the assumption that the predictions of $\Phi$ on $\mathbf{z}_1$ and $\mathbf{z}_2$ change by less than the margin value $p$. To show Condition a), observe that since $\Phi(\boldsymbol{\theta}; \mathbf{x})$ is piecewise linear in $\mathbf{x}$, $\nabla_{\boldsymbol{\theta}} \Phi(\boldsymbol{\theta}; \mathbf{x})$ is affine in $\mathbf{x}$ inside each activation pattern, and can be written as $\nabla_{\boldsymbol{\theta}} \Phi(\boldsymbol{\theta}; \mathbf{x}) = \mathbf{A}\mathbf{x} + \mathbf{c}$ for some matrix $\mathbf{A}$ and vector $\mathbf{c}$. This implies that

1. $\lambda_\beta + \lambda_\alpha = \frac{\lambda_1 \beta}{\alpha + \beta} + \frac{\lambda_1 \alpha}{\alpha + \beta} = \lambda_1$,

2. $\lambda_\beta(\mathbf{x}_1 + \alpha \boldsymbol{\nu}) + \lambda_\alpha(\mathbf{x}_1 - \beta \boldsymbol{\nu}) = (\lambda_\beta + \lambda_\alpha)\mathbf{x}_1 + \frac{\lambda_1 \beta}{\alpha + \beta} \alpha \boldsymbol{\nu} - \frac{\lambda_1 \alpha}{\alpha + \beta} \beta \boldsymbol{\nu} = \lambda_1 \mathbf{x}_1$,

which is turn shows that,

$$
\begin{aligned}
\lambda_\beta \nabla_{\boldsymbol{\theta}} \Phi(\boldsymbol{\theta}; \mathbf{z}_1) + \lambda_\alpha \nabla_{\boldsymbol{\theta}} \Phi(\boldsymbol{\theta}; \mathbf{z}_2) &= \lambda_\beta(\mathbf{A}(\mathbf{x}_1 + \alpha \boldsymbol{\nu}) + \mathbf{c}) + \lambda_\alpha(\mathbf{A}(\mathbf{x}_1 - \beta \boldsymbol{\nu}) + \mathbf{c}) \\
&= (\lambda_\beta + \lambda_\alpha)\mathbf{A}\mathbf{x}_1 + (\lambda_\alpha + \lambda_\beta)\mathbf{c} \\
&= \lambda_1(\mathbf{A}\mathbf{x}_1 + \mathbf{c}) \\
&= \lambda_1 \nabla_{\boldsymbol{\theta}} \Phi(\boldsymbol{\theta}; \mathbf{x}_1).
\end{aligned}
$$

The above implies

$$\left\|\boldsymbol{\theta} - \sum_{i \neq 1} \lambda_i y_i \nabla_{\boldsymbol{\theta}} \Phi(\boldsymbol{\theta}; \mathbf{x}_i) - \lambda_\beta \nabla_{\boldsymbol{\theta}} \Phi(\boldsymbol{\theta}; \mathbf{z}_1) - \lambda_\alpha \nabla_{\boldsymbol{\theta}} \Phi(\boldsymbol{\theta}; \mathbf{z}_2)\right\|^2 = \left\|\boldsymbol{\theta} - \sum_{i=1}^n \lambda_i y_i \nabla_{\boldsymbol{\theta}} \Phi(\boldsymbol{\theta}; \mathbf{x}_i)\right\|^2 \leq \varepsilon,$$

hence, $S'$ is also an $\varepsilon$-KKT set. $\qquad\square$

## B.2 SPLITTING DISTANCE LOWER BOUNDS

In this subsection of the appendix, we establish a bound on the distance between the original point and the new points obtained through the splitting procedure when using Lemma 7 in the almost-KKT setting. The following lemma will be useful to bound the deviations in the predictions of the model that result from splitting a data instance in a direction $\boldsymbol{\nu}$.

**Lemma 14.** *Given an $(\varepsilon, \delta)$-KKT point $\boldsymbol{\theta}$, suppose that $\boldsymbol{\nu}$ is a directional vector satisfying $\|\boldsymbol{\nu}\| = 1$ and $|\langle \mathbf{x}_i, \boldsymbol{\nu}\rangle| \leq \gamma$ for all $i$, for some $\gamma > 0$. Then, we have for any neuron $j$ that*

$$|\langle \mathbf{w}_j, \boldsymbol{\nu}\rangle| \leq \varepsilon + \gamma |v_j| \sum_{i=1}^n \lambda_i.$$

*Proof.* Compute

$$
\begin{aligned}
|\langle \mathbf{w}_j, \boldsymbol{\nu}\rangle| &= \left| \langle \mathbf{w}_j, \boldsymbol{\nu}\rangle - \left\langle v_j \sum_{i=1}^n \lambda_i y_i \sigma'_{i,j} \mathbf{x}_i, \boldsymbol{\nu} \right\rangle + \left\langle v_j \sum_{i=1}^n \lambda_i y_i \sigma'_{i,j} \mathbf{x}_i, \boldsymbol{\nu} \right\rangle \right| \\
&\leq \left| \left\langle \mathbf{w}_j - v_j \sum_{i=1}^n \lambda_i y_i \sigma'_{i,j} \mathbf{x}_i, \boldsymbol{\nu} \right\rangle \right| + \left| v_j \sum_{i=1}^n \lambda_i y_i \sigma'_{i,j} \langle \mathbf{x}_i, \boldsymbol{\nu}\rangle \right| \\
&\leq \left\| \mathbf{w}_j - v_j \sum_{i=1}^n \lambda_i y_i \sigma'_{i,j} \mathbf{x}_i \right\| \cdot \|\boldsymbol{\nu}\| + |v_j| \sum_{i=1}^n |\lambda_i y_i \sigma'_{i,j}| \cdot |\langle \mathbf{x}_i, \boldsymbol{\nu}\rangle| \\
&= \left\| \mathbf{w}_j - v_j \sum_{i=1}^n \lambda_i y_i \sigma'_{i,j} \mathbf{x}_i \right\| \cdot \|\boldsymbol{\nu}\| + |v_j| \sum_{i=1}^n \lambda_i \sigma'_{i,j} \cdot |\langle \mathbf{x}_i, \boldsymbol{\nu}\rangle| \\
&\leq \varepsilon + \gamma |v_j| \sum_{i=1}^n \lambda_i.
\end{aligned}
$$

$\qquad\square$

Equipped with the above lemma, we now turn to prove the theorems.

*Proof of Theorem 8.* The theorem follows from Lemma 7, as long as $\mathbf{x}_l + \alpha_l \boldsymbol{\nu}$ has the same activation pattern as $\mathbf{x}_l$. To this end, we need to bound $|\frac{\langle \mathbf{w}_j, \mathbf{x}_l\rangle + b_j}{\langle \mathbf{w}_j, \boldsymbol{\nu}\rangle}|$, which is the distance between $\mathbf{x}_l$ and the hyperplane induced by $c_j(\mathbf{x})$, for each neuron $c_j(\mathbf{x})$.

Denoting the signed distance between a point $\mathbf{x}$ and the hyperplane $\langle \mathbf{w}_j, \mathbf{x}\rangle + b_j = 0$ by $D_j(\mathbf{x}) = \frac{\langle \mathbf{w}_j, \mathbf{x}\rangle + b_j}{\|\mathbf{w}_j\|}$, we can rewrite

$$|\alpha_l| \geq \left| \frac{D_j(\mathbf{x}_l)\|\mathbf{w}_j\|}{\langle \mathbf{w}_j, \boldsymbol{\nu}\rangle} \right| \geq \frac{|D_j(\mathbf{x}_l)|\|\mathbf{w}_j\|}{\varepsilon + \gamma |v_j| \sum_{i=1}^n \lambda_i},$$

where the last inequality follows from Lemma 14. Namely, we can take $|\alpha_l|$ as small as

$$\min_{j \in [n]} \frac{|D_j(\mathbf{x}_l)|\|\mathbf{w}_j\|}{\varepsilon + \gamma |v_j| \sum_{i=1}^n \lambda_i}.$$

The same reasoning yields the same bound for $|\beta_l|$. $\qquad\square$

*Proof of Theorem 9.* First, let us bound $|(\Phi(\boldsymbol{\theta}; \mathbf{x}_l + \alpha_l \boldsymbol{\nu}) - \Phi(\boldsymbol{\theta}; \mathbf{x}_l)))|$ as follows

$$
\begin{aligned}
|\Phi(\boldsymbol{\theta}; \mathbf{x}_l) - \Phi(\boldsymbol{\theta}; \mathbf{x}_l + \alpha_l \boldsymbol{\nu})| &= \left| \sum_{j \in J} v_j \left[ \langle \mathbf{w}_j, \mathbf{x}_l \rangle + b_j \right]_+ - \sum_{j \in J} v_j \left[ \langle \mathbf{w}_j, \mathbf{x}_l + \alpha_l \boldsymbol{\nu} \rangle + b_j \right]_+ \right| \\
&= \left| \sum_{j \in J} v_j \left( \left[ \langle \mathbf{w}_j, \mathbf{x}_l \rangle + b_j \right]_+ - \left[ \langle \mathbf{w}_j, \mathbf{x}_l + \alpha_l \boldsymbol{\nu} \rangle + b_j \right]_+ \right) \right| \\
&\leq \sum_{j \in J} |v_j| \left| \left[ \langle \mathbf{w}_j, \mathbf{x}_l \rangle + b_j \right]_+ - \left[ \langle \mathbf{w}_j, \mathbf{x}_l + \alpha_l \boldsymbol{\nu} \rangle + b_j \right]_+ \right| \\
&\overset{*}{\leq} \sum_{j \in J} |v_j| |\langle \mathbf{w}_j, \alpha_l \boldsymbol{\nu} \rangle| = \alpha_l \sum_{j \in J} |v_j| |\langle \mathbf{w}_j, \boldsymbol{\nu} \rangle| \\
&\overset{**}{\leq} \alpha_l \sum_{j \in J} |v_j| \left( \varepsilon + \gamma |v_j| \sum_{i=1}^{n} \lambda_i \right),
\end{aligned}
\tag{9}
$$

where $*$ is by the fact that $[\cdot]_+$ is 1-Lipschitz, and $**$ is by Lemma 14. If the above quantity does not deviate beyond the value of the margin $p$, then we can guarantee that Condition b) still holds and we can use Lemma 7. It now only remains to bound by how much $\delta$ deteriorates as follows

$$
\begin{aligned}
\left| \frac{\lambda_l \beta_l}{\alpha_l + \beta_l} (y_l \cdot \Phi(\boldsymbol{\theta}; \mathbf{x}_l + \alpha_l \boldsymbol{\nu}) - p) \right| &= \left| \frac{\lambda_l \beta_l}{\alpha_l + \beta_l} (y_l \cdot \Phi(\boldsymbol{\theta}; \mathbf{x}_l + \alpha_l \boldsymbol{\nu}) - y_l \cdot \Phi(\boldsymbol{\theta}; \mathbf{x}_l) + y_l \cdot \Phi(\boldsymbol{\theta}; \mathbf{x}_l) - p) \right| \\
&= \left| \frac{\lambda_l \beta_l}{\alpha_l + \beta_l} (y_l \Phi(\boldsymbol{\theta}; \mathbf{x}_l) - p) + \frac{\lambda_l \beta_l}{\alpha_l + \beta_l} (y_l (\Phi(\boldsymbol{\theta}; \mathbf{x}_l + \alpha_l \boldsymbol{\nu}) - \Phi(\boldsymbol{\theta}; \mathbf{x}_l))) \right| \\
&\leq \left| \frac{\lambda_l \beta_l}{\alpha_l + \beta_l} (y_l \Phi(\boldsymbol{\theta}; \mathbf{x}_l) - p) \right| + \left| \frac{\lambda_l \beta_l}{\alpha_l + \beta_l} (y_l (\Phi(\boldsymbol{\theta}; \mathbf{x}_l + \alpha_l \boldsymbol{\nu}) - \Phi(\boldsymbol{\theta}; \mathbf{x}_l))) \right| \\
&\leq \frac{\beta_l}{\alpha_l + \beta_l} \delta + \frac{\lambda_l \beta_l}{\alpha_l + \beta_l} |y_l| |(\Phi(\boldsymbol{\theta}; \mathbf{x}_l + \alpha_l \boldsymbol{\nu}) - \Phi(\boldsymbol{\theta}; \mathbf{x}_l)))| \\
&= \frac{\beta_l}{\alpha_l + \beta_l} \delta + \frac{\lambda_l \beta_l}{\alpha_l + \beta_l} |(\Phi(\boldsymbol{\theta}; \mathbf{x}_l + \alpha_l \boldsymbol{\nu}) - \Phi(\boldsymbol{\theta}; \mathbf{x}_l)))| \\
&\leq \delta + \lambda_l \alpha_l \sum_{j \in J} |v_j| \left( \varepsilon + \gamma |v_j| \sum_{i=1}^{n} \lambda_i \right),
\end{aligned}
$$

where the last inequality is due to Equation 9 and the fact that $\frac{\beta_l}{\alpha_l + \beta_l} \leq 1$. The theorem then follows by summing the above bound with an analogous bound obtained for $\beta_l$. $\qquad \square$

## C   FINDING AN ALMOST ORTHOGONAL SPLITTING DIRECTION USING SVD DECOMPOSITION

Theorems 8 and 9 require finding a direction $\boldsymbol{\nu}$ such that $\langle \boldsymbol{\nu}, \mathbf{x}_i \rangle$ for all data instances $\mathbf{x}_i$ to be effective. Conveniently, such an upper bound can be derived in terms of the smallest singular value of the data matrix. Given a training set $S$, we write it as a column matrix $\mathbf{S} = (\mathbf{x}_1, \ldots, \mathbf{x}_n)$. Then we have the following upper bound on the dot products.

**Theorem 15.** *Let $\mathbf{S} = \mathbf{U} \boldsymbol{\Sigma} \mathbf{V}^\top$ be the SVD decomposition of the training data matrix $\mathbf{S}$, where $\mathbf{U} \in \mathbb{R}^{d \times d}, \mathbf{V} \in \mathbb{R}^{n \times n}$, and $\boldsymbol{\Sigma} \in \mathbb{R}^{d \times n}$ is a diagonal matrix with entries $\sigma_1 \geq \ldots \geq \sigma_d \geq 0$. Then, there exists a unit vector $\boldsymbol{\nu}$ such that $|\langle \mathbf{x}_i, \boldsymbol{\nu} \rangle| \leq \sigma_d$ for all $i \in [n]$.*

*Proof.* Let us chose $\boldsymbol{\nu} = \mathbf{U}_d$, i.e the $d$-th column in $\mathbf{U}$. Let us denote $\mathbf{X}_i$ to be the $i$-th column. We need to show that $|\langle \mathbf{X}_i, \boldsymbol{\nu} \rangle| \leq \sigma_d$, or equivalently $|\mathbf{X}_i^\top \boldsymbol{\nu}| \leq \sigma_d$. We can write $\mathbf{X}_i$ as $\mathbf{X}_i = \mathbf{U} \boldsymbol{\Sigma} \mathbf{V}_i^\top = \sum_{j=1}^{d} \sigma_j \mathbf{V}_{i,j}^\top \mathbf{U}_j$ where $\mathbf{V}_{i,j}^\top$ is the $j$-th entry of the vector $\mathbf{V}_i$. Compute

$$
|\langle \mathbf{x}_i, \boldsymbol{\nu} \rangle| = \left| \left\langle \sum_{j=1}^{d} \sigma_i \mathbf{V}_{i,j}^\top \mathbf{U}_i, \mathbf{U}_d \right\rangle \right| = \left| \sum_{j=1}^{d} \sigma_i \mathbf{V}_{i,j}^\top \langle \mathbf{U}_i, \mathbf{U}_d \rangle \right| \overset{*}{=} |\sigma_d \mathbf{V}_{i,d}^\top| \overset{**}{\leq} \sigma_d,
$$

where $*$ is due to the fact that $\mathbf{U}$ is orthogonal and $**$ is due to the fact that $\mathbf{V}$ is also orthogonal, and $\sigma_d \geq 0$. □

We remark that the above immediately implies that $\gamma \leq \sigma_d$ for $\gamma$ in the statements of Theorems 8 and 9. Note that the final inequality may be rather loose, since $\mathbf{V}_d^\top$ is a unit vector; equality holds if and only if the entry $\mathbf{V}_{i,d}^\top$ equals 1. If, however, the vector $\mathbf{V}_{i,d}^\top$ has more equally distributed entries, a tighter upper bound with a magnitude of roughly $\sigma_d/n$ will hold, further amplifying the efficacy of our theorems.

## D  SHIFTING THE TRAINING SET POST-TRAINING

In this appendix, we provide additional details on how we obtained the trained network for our CIFAR experiment in Section 4. Recall that our goal was to shift the training data by a constant bias and then retrain the architecture used by Haim et al. (2022) on the shifted data, to obscure the attacker's prior knowledge. Since shifting the training set by a constant bias shifts the gradients of the resulting objective function by the same bias, shifting both the data and the initialization point leads to convergence to the same neural network as obtained by shifting the network trained on the original, unshifted data.

In light of this, rather than shifting the data and retraining the network from scratch, we utilized the network trained by Haim et al. (2022) and adjusted its biases in the first hidden layer. As an example, consider a hidden neuron with weights $\mathbf{w}$ and bias $b$. Given an input $\mathbf{x}$, the pre-activation output of this neuron is $\langle \mathbf{w}, \mathbf{x} \rangle + b$. Suppose we wish to shift each coordinate of the input $\mathbf{x}$ by a vector $\mathbf{u}$. In this case, the pre-activation output of the shifted input is

$$\langle \mathbf{w}, \mathbf{x} + \mathbf{u} \rangle + b = \langle \mathbf{w}, \mathbf{x} \rangle + \langle \mathbf{w}, \mathbf{u} \rangle + b.$$

Since $\langle \mathbf{w}, \mathbf{u} \rangle$ is constant, by modifying the bias term via the transformation $b \mapsto b - \langle \mathbf{w}, \mathbf{u} \rangle$, we recover the behavior of the original network on the shifted input. Applying this transformation to all biases in the first hidden layer for various values of $\mathbf{u}$, we obtained the networks used in Figure 2b.

## E  ON THE SIZE OF THE TRAINING SET

In this section, we examine how the attacker's knowledge of the number of training points influences our analysis. The size of the training set can act as a prior and can constrain the applicability of Lemma 2 and Lemma 3. Moreover, we view this information as a strong prior for an attacker to possess. Yet, even if the attack does have this number it doesn't necessarily mean they can reconstruct the training data. First, the lemmas can be used in a manner that preserves the total number of samples. For example, if two points are split and then points from different splits are merged, we obtain a different set of the same size. Second, even if our analysis is limited, this does not mean there are no infinitely many global minima achieved by sets of the same size. We don't know how strong this prior is.

## F  MORE EXPERIMENTS ON SHIFTING THE TRAINING DATA

In this section we conduct more experiments, covering more training sets and more architectures. We experienced consistent results with the experiments in 4 which strengthen our claim that a prior plays an important role in the attack.

### F.1  SPHERE WITH SMALL RADII

We expended the experiment presented in Figure 2a to include small radii. The network was trained with training points from the unit sphere. Then we attacked it with initializations from the spheres of radii 0.1, 0.5 and 1. As shown in Figure 3 initializations with the smaller radii give better results because in high dimension the points on the unit sphere are very far from each other, so sampling points from a sphere with a smaller radius actually gives us initialization points that are closer to the training points.

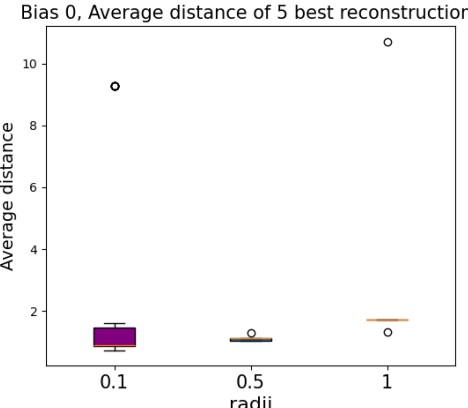

Figure 3: A comparison of reconstruction results based on initializations with small radii, for a network that was trained on data sampled from the unit sphere in $\mathbb{R}^{250}$ and measured by the average Euclidean distance of the 5 best reconstructions. Initializations with the smaller radii give better results because in high dimension the points on the unit sphere are very far from each other, so sampling points from a sphere with a smaller radius actually gives us initialization points that are closer to the training points.

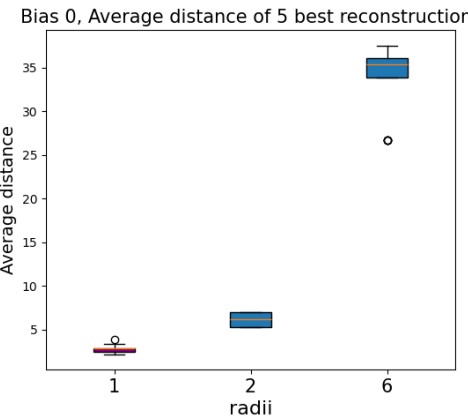

Figure 4: A comparison of reconstruction results obtained from initializations with varying radii is presented for a network trained on data drawn from an annular region with radii in the range 1-4. Performance is evaluated using the average Euclidean distance of the 5 best reconstructions. The $x$-axis indicates the minimum radius of the attack's initialization (with the maximum radius fixed at $\min + 3$). As the initialization radius deviates further from the training distribution, reconstruction quality consistently degrades.

### F.2 ANNULUS EXPERIMENT

We constructed a dataset whose points lie in an annulus: all examples have norms between radii 1 and 4 (chosen arbitrarily) and are drawn from this ring (support). We then ran the reconstruction attack multiple times with different initializations (priors): Initialization with radii between 1 and 4 (matching the support of the true distribution), Initialization with radii between 2 and 5 (partially overlapping support). Initialization with radii between 6 and 9 (no overlap with the true support). We measured the average distance of the 5 best reconstructions from the original training set. The results of this experiment show that the greater the overlap between the support and the support of the training distribution, the better the attack performed. The results are shown in Figure 4.

### F.3 VARIOUS DATASETS

In this subsection, we tested the training shift procedure on MNIST (Figure 5) and ImageNet (Figure 6).

In the MNIST experiment, we trained two fully connected neural networks: one with all images' pixels shifted by 0.1, and one with all images' pixels shifted by 5. The figure shows that while we could reconstruct some images when shifting by 0.1, we could not construct even a single image when shifting by 5.

When training on ImageNet, we shifted the pixels by 0.01 and by 0.1. Once again, we see that the quality of the reconstruction depends on the extent to which the images are shifted.

We observe consistent results from the experiments in Section 4: the greater the data shift, the less likely the attack is to succeed. This strengthens our claim that shifting the data can mask the prior.

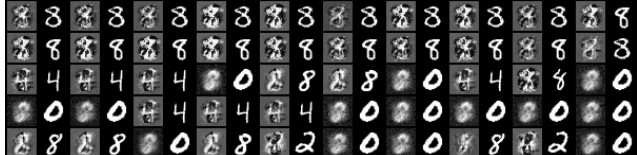

(a) Reconstructed images (odd columns) and their nearest neighbors from the training set (even columns). All pixels were shifted by 0.1.

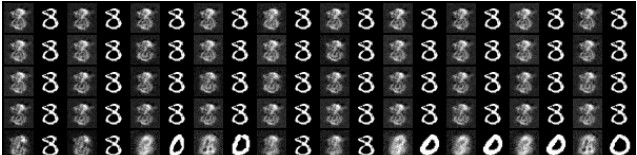

(b) Reconstructed images (odd columns) and their nearest neighbors from the training set (even columns). All pixels were shifted by 5.

Figure 5: Reconstructed MNIST images (odd columns) and their nearest neighbors from the training set (even columns). In the top image, all training data pixels were shifted by 0.1, whereas in the bottom image, they were shifted by 5. We trained a model on the shifted data until it reached an almost-KKT point, without any regularization. The experiment demonstrates that as the data is shifted further, corresponding to a weaker prior available to the attacker, the effectiveness of the attack diminishes rapidly. While the top reconstruction still captures vague characteristics of a small subset of the training set, the bottom reconstruction fails entirely.

### F.4 MULTICLASS CLASSIFIER

In this subsection, we check if the shift mechanism holds for multiclass scenario. We trained a network on CIFAR10, where its first layer is a convolution layer. We trained it once where we moved all the pixels by 0.01 and once where we moved all the pixels by 0.1 ((Figure 8). As the figure shows, the weaker the prior the attacker has, the less likely it could reconstruct the training samples.

### F.5 EARLY STOP

In this subsection, we repeated the sphere experiment in Figure 2a, but this time stopped the attack once it reached a KKT loss of 333. Since different attacks yield different KKT losses, the results may be affected directly by this. We wanted to neutralize this effect. As can be seen in Figure 7, the results are similar to those in Figure 2a, suggesting that the difference in KKT loss between attacks did not affect the quality of reconstruction.

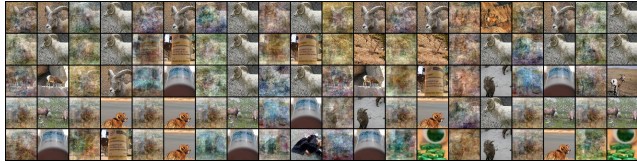

(a) Reconstructed images (odd columns) and their nearest neighbors from the training set (even columns). None of the pixels were shifted.

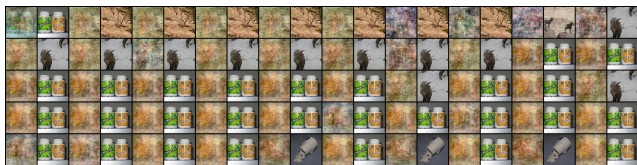

(b) Reconstructed images (odd columns) and their nearest neighbors from the training set (even columns). All pixels were shifted by 0.5.

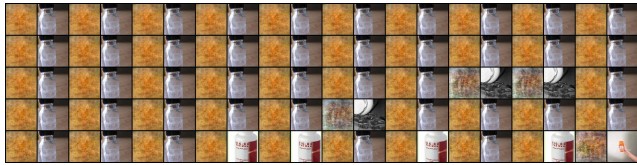

(c) Reconstructed images (odd columns) and their nearest neighbors from the training set (even columns). All pixels were shifted by 5.

Figure 6: Reconstructed ImageNet images (odd columns) and their nearest neighbors from the training set (even columns). In the middle image, all training data pixels were shifted by 0.5, and in the bottom image by 5. We trained a model on the shifted data until it reached an almost-KKT point, without any regularization. The experiment demonstrates that as the data is shifted further, corresponding to a weaker prior available to the attacker, the effectiveness of the attack diminishes rapidly. While the middle reconstruction still captures vague characteristics of a small subset of the training set, the bottom reconstruction fails entirely. The top image is a baseline as there is no baseline in Haim et al. (2022) for ImageNet.

## G    DISCLOSURE OF LLM USAGE

Parts of this manuscript were polished for clarity and style using OpenAI's ChatGPT (GPT-5). The model was used solely for language refinement and proofreading; all research ideas, technical content, derivations, and conclusions are the authors' own.

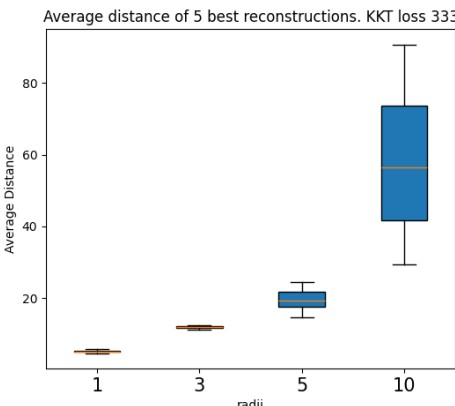

Figure 7: A comparison of reconstruction outcomes obtained using different initialization radii, where each attack is early-stopped once the loss reaches 333. The network was trained on data sampled from the unit sphere, and performance was measured as the average Euclidean distance among the five best reconstructions. The radius increases as the prior knowledge available to the attacker weakens, significantly degrading the quality of the reconstruction.

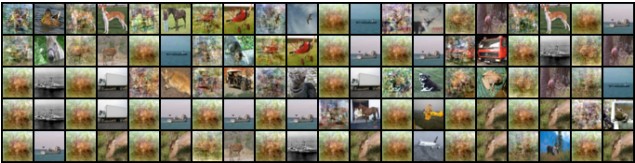

(a) Reconstructed images (odd columns) and their nearest neighbors from the training set (even columns). All pixels were shifted by 0.01.

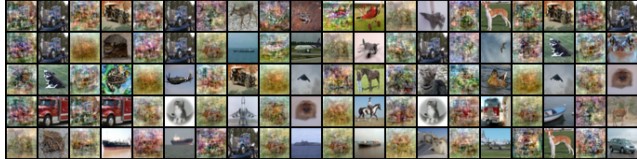

(b) Reconstructed images (odd columns) and their nearest neighbors from the training set (even columns). All pixels were shifted by 0.1.

Figure 8: Reconstructed CIFAR10 images (odd columns) and their nearest neighbors from the training set (even columns) in the multiclass scenario. In the top image, all training data pixels were shifted by 0.01, and in the bottom image by 0.1. The experiment demonstrates that even in the multiclass scenario, as the data is shifted further, corresponding to a weaker prior available to the attacker, the effectiveness of the attack diminishes rapidly. This suggests that our analysis for the binary case .

