# OpenReview forum: "No Prior, No Leakage: Revisiting Reconstruction Attacks in Trained Neural Networks"
_ICLR.cc/2026/Conference — ICLR 2026 Poster_

### Official Review · Reviewer_1FR8 · 2025-10-29

**Soundness:** 3
**Presentation:** 3
**Contribution:** 3
**Rating:** 4
**Confidence:** 4

**Summary:**

This paper revisits reconstruction attacks based on implicit bias (e.g., KKT-condition–based attacks) and provides a rigorous theoretical analysis of their limitations. The authors show that without prior knowledge about the data distribution, such reconstruction objectives admit infinitely many global optima that are arbitrarily far from the true training samples. The paper also introduces the notion of a “secret bias,” illustrating that hidden transformations or priors can effectively protect data from being reconstructed.

**Strengths:**

- **Solid theoretical foundations.**
  The mathematical analysis is thorough and clearly presented. The authors formalize the implicit assumptions in prior KKT-based attacks and prove that, without priors, the reconstruction loss function has infinitely many degenerate global minima. This provides a precise theoretical characterization of why such attacks cannot succeed “from scratch.”

- **Novel perspective on priors and implicit bias.**
  Instead of proposing yet another reconstruction algorithm, the paper takes a principled view, showing that the presence (or absence) of prior knowledge fundamentally determines whether data can be reconstructed. This shifts the discussion of privacy leakage from algorithmic ingenuity to epistemic limitations.

**Weaknesses:**

1. **Limited empirical validation.**
   The experimental results are somewhat narrow, being limited to a shallow 3-layer network on the CIFAR dataset. To strengthen the empirical evidence, it would be helpful to include additional datasets (e.g., MNIST, CelebA, ImageNet) and deeper or modern architectures (e.g., ResNet, ViT) to confirm that the theoretical claims hold in more realistic settings.

2. **Lack of quantitative evaluation of prior strength.**
   The paper emphasizes that both the “secret bias” and the attacker’s prior knowledge are key to reconstruction success, but it does not measure how the degree or accuracy of prior information affects attack outcomes.
   A systematic experiment varying the strength of prior knowledge (e.g., partial mean shift known, partial pixel-range constraint, etc.) and plotting reconstruction quality or KKT loss would make the argument more convincing.

**Questions:**

1. Can the theoretical framework extend beyond classification to regression or generative models (e.g., diffusion models)?
2. Could the “secret bias” defense be generalized or implemented in practical training pipelines without harming model performance?

---

> ### Author Response · Authors · 2025-11-24
>
> We thank the reviewer for their time they put into the review. We appreciate their recognition of our theoretical foundation and the novel perspective on the role of priors in KKT-based attacks. We address all the comments raised by the reviewer below:
>
> W1. **Limited empirical validation** To strengthen the empirical validations, we added the following experiments:
>
> **MNIST experiment:** We trained two fully connected neural networks, one with all training images’ pixels shifted by 0.1 and one with all pixels shifted by 5 (Figure 5, Appendix F). While we could still reconstruct some images in the first scenario, we could not reconstruct any in the second. In addition, we conducted more experiments to check if the claim holds for broader settings:
>
> **ImageNet experiment:** We trained two fully-connected neural networks - the first network was trained on images where all pixels were shifted by 0.01, and the second one was trained on images where they were shifted by 0.1 (Figure 6, Appendix F). While we could reconstruct some images in the first scenario, we could not reconstruct any in the second.
>
> **Multiclass experiment:** we trained two multiclass classifiers on CIFAR-10, where the first layer is a convolution layer (CNN): one on images whose pixels were shifted by 0.01, and another on images whose pixels were shifted by 0.1. We then attacked these models using the multiclass KKT-based method of (Buzaglo et. al. 2023), i.e., by minimizing the multiclass KKT loss. The results (Figure 8, Appendix F) show that the reconstruction quality of the network trained with a shift of 0.01 is higher than that of the network trained with a shift of 0.1, empirically supporting our claim that, in the multiclass setting, a sufficiently strong prior is necessary to reconstruct the training examples.
> All experiments empirically support our claim that a stronger prior implies better reconstruction quality.
> Finally, we conducted an experiment on ResNet architecture with two and three ResNet classification blocks, using CIFAR-10. When attacking the network, we were unable to reconstruct any image. This is because the network did not converge to a KKT point during training. We stress that even though ResNet is not homogeneous, this does not necessarily mean it cannot reach a KKT point, and other ResNet architectures may.
>
> W2. **Evaluation of the prior:** We appreciate the reviewer’s attention to the importance of quantitatively assessing the prior's strength. As shown in Figure 2(a), we conducted such an evaluation by training a neural network on data sampled from the unit sphere and then repeatedly performing the attack with different initialization radii. The box plot demonstrates that when the initialization is farther from the unit sphere, the reconstructed samples deviate more from the true training example. This trend aligns well with the role of the prior emphasized in the review.
> In addition, we added the following experiment:
>
> **Annulus experiment:** We constructed a dataset whose points lie uniformly in an annulus (Figure 4, Appendix F): all examples have norms between radii 1 and 4 (chosen arbitrarily) and are drawn from this ring (support).
> We then ran the reconstruction attack multiple times with different initializations (priors):
> Initialization with radii between 1 and 4 (matching the support of the true distribution),
> Initialization with radii between 2 and 5 (partially overlapping support).
> Initialization with radii between 6 and 9 (no overlap with the true support).
> We measured the average distance of the 5  best reconstructions from the original training set. The results of this experiment show that the greater the overlap between the support and the support of the training distribution, the better the attack performed.
>
> As shown in the new experimental results in Appendix F of the revised version, reconstruction quality improves as the prior’s support overlaps more with the true data distribution.

---

> > ### Author Response · Authors · 2025-11-24
> >
> > Q1. **Beyond Classification:** This is an important question. While we don’t know whether our theoretical framework can be extended to regression or generative models as there is no known way to make a non-homogeneous network converge to a KKT point, we believe it can be extended to other cases, such as multiclass classification based on the multiclass experiment  (Figure 8, Appendix F).
> >
> > Q2.**Secret bias training pipeline:**  As for implementing the idea of “secret bias” in the training pipeline, it is a valid and practical idea. The bias can be incorporated into the training pipeline by first choosing and storing a fixed bias, then shifting all training samples accordingly. After initializing the network, its weights are shifted by the same bias before training proceeds using any a gradient-based method. During inference, each input sample is similarly shifted by the stored bias before being fed into the network.
> > Keep in mind that the bias is secret and must not fall into an attacker’s hands.
> >
> > This explanation also appears in Appendix D.

---

> > > ### Comment · Area_Chair_uQYi · 2025-11-26
> > > **Interaction needed**
> > >
> > > Dear Reviewer,
> > > Thanks for your time and effort in reviewing ICLR2026 submissions. The authors have provided their responses to your reviews. Please read and raise your further comments, and discuss with the authors.
> > > Best regards,
> > > Your AC

---

### Official Review · Reviewer_2UcW · 2025-10-31

**Soundness:** 2
**Presentation:** 3
**Contribution:** 2
**Rating:** 4
**Confidence:** 3

**Summary:**

- This paper studies the attack presented in [1], specifically focused on the importance of the prior knowledge loss term in the reconstruction loss
-This loss encodes information about the pixel min/max values, and constraints the reconstructed images to be within this valid set
- Theoretically, they show that through two operations: splitting and merging, it is possible to construct sets which satisfy the reconstruction loss exactly (excluding the prior term), but are not the original dataset
- They extend this analysis to cases where the reconstruction loss is only fit within a error term
- Empirically they show that when the image initialization is mismatched with the statistics of the actual images, reconstruction quality suffers dramatically





[1] Reconstructing training data from trained neural networks

**Strengths:**

- Presentation is solid, idea is well motivated
- Provides some theoretical understanding into these reconstruction losses, which generally speaking, lags far behind the empirical studies of them

**Weaknesses:**

- Lemma 2 and 3 require that the split/merged datapoints match the activation patterns of the original datapoints. It is unclear how much of a restriction this is. Can the authors empirically demonstrate that such operations are possible (on slightly less toy data than the 2d example presented). While it seems interesting, I am unconvinced that it is possible, for example, to split a datapoint with any reasonably large \alpha, \beta splitting coefficients, so that the split datapoints are actually "far" from the original. While the authors claim that reconstructed can be "arbitrarily far" from the original dataset, it seems this activation pattern constraint could actually restrict is substantially.

- Empirical evidence is somewhat unconvincing. From experience, the attack presented in [1] generally is very brittle and hard to get working. Indeed, this paper aims to study why this might be the case. The authors present empirical evidence on CIFAR-100, where the attack already doesn't work very well. It would be additionally be useful if the authors present results on MNIST (or a simpler dataset), where the attack works better, so that there can be a greater difference between the version of the attack without prior knowledge/with prior knowledge.


[1] Reconstructing training data from trained neural networks

**Questions:**

- The authors report that all runs achieve a similar KKT loss of 330-332. If I remember correctly, the KKT loss of this attack does not actually decrease significantly over the course of training, so this gap of 330/332 may actual be significant. I would suggest setting up this experiment differently. Instead, would it be possible to terminate all runs when they achieve the same KKT loss exactly? I.e. do early stopping on all runs when they hit loss = 333.0 exactly, for example, then compare quality. This would prevent the issue where the remaining 0-2 loss points carry the majority of image quality.

- Empirically, what KKT loss does the original dataset achieve? I.e. if we initialize the reconstruction images with the original dataset, and compute the loss, what is the result? Is it below 330-332?

---

> ### Author Response · Authors · 2025-11-24
>
> We thank the reviewer for their time and effort spent on the review. We appreciate the acknowledgment of our theoretical analysis.
>
> W1. **Usability of Lemmas** Regarding Lemmas 2 and 3 and the activation patterns: the purpose of these lemmas is to show that the objective function used in the attack devised by Haim et al. has multiple minima, rather than a single solution corresponding to the training set. Specifically, in Theorem 4, we have rigorously proved that when the training data does not span the entire domain, we can split arbitrarily far from the training set (i.e., $\alpha$ and $\beta$ can be arbitrarily large). In Theorem 8, we analyzed a more realistic setting in which the training data spans the entire domain and showed lower bounds on how far we can split. While this still imposes some quantitative limitations on how far we can split, it qualitatively shows that well-trained models closer to KKT are more resilient to this privacy attack. Moreover, if the data is well-structured in the sense that it lies approximately on a lower-dimensional manifold, this increases our splitting budget, thereby strengthening the applicability of our results under suitable distributional assumptions. We would like to stress that a core part of the proof is showing that the new points have the same activation pattern as the original point, indicating that this is indeed possible. Empirical evidence is presented in all our experiments, specifically in Figure 2(a), where we show that even when the attack achieves a KKT loss similar to that of the real training set, it produces samples that are far from the training samples. We are currently running an additional experiment to validate empirically how far points can be split and will update shortly with our findings.
>
> W2. **Empirical Evidence** Following the reviewer's interesting suggestion, we have added an MNIST experiment in which the attack may work better.
>
> **MNIST experiment:** We trained two depth 3, width 1000 fully connected neural networks, one with all training images’ pixels shifted by 0.1 and one with all pixels shifted by 5. While we could still reconstruct some images in the first scenario, we could not reconstruct any in the second (Figure 5, Appendix F).
>
> **Annulus experiment:** We added another synthetic dataset to make the attack even more successful. We constructed a dataset whose points lie in an annulus: all examples have norms between radii 1 and 4 (chosen arbitrarily) and are drawn from this ring (support). We then ran the reconstruction attack multiple times with different initializations (priors):
> Initialization with radii between 1 and 4 (matching the support of the true distribution),
> Initialization with radii between 2 and 5 (partially overlapping support).
> Initialization with radii between 6 and 9 (no overlap with the true support).
> We measured the average distance of the 5  best reconstructions from the original training set. The results of this experiment show that the greater the overlap between the support and the support of the training distribution, the better the attack performed (Figure 4, Appendix F).
>
> In addition, we conducted more experiments to check if the claim holds for other datasets:
> We ran an additional experiment on ImageNet, training a fully connected neural network twice: once on images with all pixels shifted by 0.01 and once on images with all pixels shifted by 0.1. Once again, while we could reconstruct some images in the first scenario, we could not reconstruct any in the second.
>
> To evaluate in even broader settings, we added the following experiment on multiclass classification: we trained two multiclass classifiers on CIFAR-10, with the first layer a convolutional layer (CNN), one on images whose pixels were shifted by 0.01, and another on images whose pixels were shifted by 0.1. We then attacked these models using the multiclass KKT-based method presented in Buzaglo et al., i.e., by minimizing the multiclass KKT loss. The results show that the reconstruction quality of the network trained with a shift of 0.01 is higher than that of the network trained with a shift of 0.1, empirically supporting our claim that, in the multiclass setting, a sufficiently strong prior is necessary to reconstruct the training examples.
> All experiments empirically support our claim that a stronger prior implies better reconstruction quality. All details are presented in Appendix F.
> Finally, we conducted an experiment on the ResNet architecture with two and three ResNet classification blocks on CIFAR-10. When attacking the network, we were unable to reconstruct any image. This is because the network did not converge to a KKT point during training. Even though ResNet is not homogeneous, this does not necessarily imply that it cannot reach a KKT point under suitable conditions, and other ResNet architectures may do so.

---

> > ### Author Response · Authors · 2025-11-24
> >
> > Q1.**KKT loss gap** Regarding the point at which all runs achieve similar loss, the loss decreases significantly during the attack. It starts with 376. Nevertheless, we added an additional experiment with an early stop at 333 (Figure 7, Appendix F). The results remain the same, indicating that the gap between 330 and 332 does not affect the results.
> >
> > Q2. **KKT loss in original attack** We reproduced the experiment in Haim et al., and the real dataset leads to a KKT loss of 331. Since this is roughly the same KKT loss achieved in our reconstruction experiments when the data are shifted, it is evident that we achieve similar values in both regimes.

---

### Official Review · Reviewer_3CEM · 2025-10-31

**Soundness:** 3
**Presentation:** 3
**Contribution:** 3
**Rating:** 8
**Confidence:** 4

**Summary:**

The paper provides theoretical analysis of the KKT-based data reconstruction objective in the seminal work Haim et. al., and showed that perhaps surprisingly, if one throw away the regularization term based on prior information in the reconstruction objective, then one can construct new data records that are global optima for the reconstruction objective, yet are infinitely far away from the training dataset in Euclidean distance. Their construction is simple yet intuitive, linear combinations of records on the margin that have the same label and same activation pattern would still lie in the margin, and moving training data records along the margin would not break the KKT condition. They then prove that when the training data does not span the full space, one can construct data records with the same label and same activation pattern of a training data that lie arbitrarily far in Euclidean distance. The assumption could be further relaxed to training data that only approximately falls into a low-dimensional space, under which the maximum feasible perturbation scale (without breaking KKT) is inverse proportional to the smallest singular value of the data matrix S. They then extend their analysis to the approximate training case, where the trained model may be $\delta$ away from the KKT condition, and prove that interestingly, if the adversary does not know $\delta$, similar splitting constructions still hold; but if the adversary knows $\delta$, then the feasible perturbation scale of the splitting construction significantly decreases. Experiments for ReLU networks on synthetically constructed classification dataset, as well as shifted CIFAR10 dataset, confirm that the reconstructed image's similarity to true training data degrades with increasing shift (corresponding to weaker prior knowledge).

**Strengths:**

The paper is well-written, and performs thorough theoretical analysis of the reconstruction objective.

The construction of "non-training data" that satisfy KKT condition is intuitive and interesting. The further extended analysis for models that only satisfy approximate KKT condition is non-trivial and aligns well with experiments.

**Weaknesses:**

The only limitation I see is that the analysis and experiments seem restricted to homogeneous networks, while most recent NN models are non-homogeneous. I understand this limitation is inherent for KKT-based reconstruction objectives, but it would be nice if the authors could discuss or use experiments to understand whether the insights generalize to typical other NN models, such as CNN.

**Questions:**

Besides weakness, I have a followup question on prior knowledge. The construction relies on splitting, or merging training data records, while in practice the adversary may be stronger prior knowledge on the number of training data records. In such case, it appears that possibly, a large fraction of the reconstructed data could still be the true training data. This could be another layer of prior knowledge worth discussing, besides the initialization.

---

> ### Author Response · Authors · 2025-11-24
>
> We thank the reviewer for their time and thoughtful feedback. We appreciate the positive assessment of our theoretical analysis and constructions, and find the question about the number of training points very constructive.
>
> W1. **CNN architecture** First, the class of homogeneous networks already includes many standard models: convolutional networks (CNN), MLPs with ReLU/Leaky ReLU activations, and pooling layers without biases, etc. Architectures, such as transformers and softmax layers, are not homogeneous. *However*, the attack in Haim et al. (and thus by extension also our analysis) is formulated at the level of the KKT conditions, and in principle applies to any network whose trained parameters satisfy (or approximately satisfy) these KKT-style stationarity relations, whether homogeneous or not. Formal convergence guarantees for such KKT points are currently known mainly for homogeneous networks (Lyu and Li, 2020; Ji and Telgarsky, 2020). Nonetheless, limitations of our theoretical understanding of the extent of convergence to KKT conditions in more general settings do not mean that other architectures cannot empirically reach similar regimes.
> To experimentally illustrate that our insights extend beyond the simplest fully-connected/binary setting, we added a multiclass CNN experiment (Figure 8, Appendix F.) There, we train two multiclass classifiers on CIFAR-10 with a convolutional first layer: one on images whose pixels are shifted by 0.01, and another on images shifted by 0.1. We then attack both models using the multiclass KKT-based method of (Buzaglo et. al 2023), i.e., by minimizing the multiclass KKT loss. The results show that the reconstruction quality is higher for the network trained with a shift of 0.01 than for the one trained with a shift of 0.1, empirically supporting our claim that, in the multiclass setting, a sufficiently strong and well-aligned prior is necessary for successful reconstruction, and that our conclusions are relevant for CNN-based models as well.
>
> Q1. **Number of training points** The reviewer is correct in their observation that the number of training points itself serves as a prior, and knowing the size of the training set can limit the effectiveness of our analysis. For this reason, in our paper, we assume that the attacker does not know the number of training points (when presenting the framework in line 216, and in remark 1), similarly to the assumptions made in Haim et al. In addition, we added the following discussion in Appendix E: “In this section, we examine how the attacker's knowledge of the number of training points influences our analysis. The size of the training set can act as a prior and can constrain the applicability of the merge and split lemmas. Moreover, we view this information as a strong prior for an attacker to possess. Yet, even if the attack does have this number, it doesn't necessarily mean they can reconstruct the training data. First, the lemmas can be used in a manner that preserves the total number of samples. For example, if two points are split and then points from different splits are merged, we obtain a different set of the same size. Second, even if our analysis is limited, this does not mean there are no infinitely many global minima achieved by sets of the same size. We don't know how strong this prior is.”

---

### Official Review · Reviewer_e1G7 · 2025-11-01

**Soundness:** 3
**Presentation:** 3
**Contribution:** 3
**Rating:** 6
**Confidence:** 4

**Summary:**

This work considers parameter-only reconstruction attacks against shallow homogeneous ReLU neural networks performing binary classification, specifically analyzing the importance of the prior in the reconstruction attack developed in (Haim et al. 2022). The main contributions are the following:

In the event where a model converges to a KKT point as described by (Lyu and Li (2020), Ji and Telgarsky (2020)), the paper describes a family of candidate training sets indistinguishable to an adversary using the (Haim et al. 2022) attack without prior information through techniques of merging and splitting training data examples. The paper describes how, under an assumption that the training data does not span the whole input space, the adversary can find training sets arbitrarily far from the original training set. The paper also analyzes how far candidate training sets can be from the original training set under a relaxed assumption of closeness of the training data to some linear subspace.

In the event where a model converges to a $(\varepsilon, \delta)$-KKT point (an approximate KKT point), the paper proves the effectiveness of similar merging and splitting techniques, with splitting at the cost of a worse $\delta$ value. The paper explores how large the family of split training sets can be depending on the adversary's knowledge of $\delta$, suggesting that models that have converged more completely may be more difficult to attack.

The paper supports its theoretical arguments with empirical results on 1) synthetic data generated along a hypersphere and 2) CIFAR, showing how a lack of prior on training data magnitude can lead to poor reconstructions, suggesting a possible defense method against such attacks (manually shifting all samples by some magnitude).

**Strengths:**

The paper is well-organized and easy to follow.

The paper adds clear theoretical analysis to the rapidly expanding discussion on the importance of training data priors in reconstruction attacks (see e.g. [Chaudhuri et al. 2025]).

The paper expands the collective understanding of the conditions necessary for the (Haim et al. 2022) attack to be useful.

The empirical results clearly support the theoretical analysis, particularly in the 'interpolated' training examples shown in Figure 2b.

**Weaknesses:**

In the empirical results, the discussion of prior information available to the adversary is limited to data magnitude; while the plausibility of other forms of prior information, such as knowledge of the training data geometry or knowledge of the KKT $\delta$, are discussed in the theoretical analysis, these are not mentioned in the empirical analysis.

The discussion could benefit from an expanded set of suggestions for future work in this field, including the potential to expand these results to the larger, higher-capacity models that are often used in empirical analyses of data reconstruction (see e.g. Struppek et al. 2022).

The empirical results and/or discussion could benefit from an analysis of potential adaptive attacks, in which a defender is aware that the magnitudes of the training data may have been altered.

Specific areas for clarification / improvement:

- In Equation (6), why are the arguments of $L_{stationary}$ only $(x_1, ..., x_l)$, as opposed to $(x_1, ..., x_l, \lambda_1, ..., \lambda_l)?$

- In the synthetic data experiment, why are no radii less than 1 tested? Would such a test be plausible, as a version of the adversary enforcing an overly strong prior on the data?

**Questions:**

Do we have any guarantees on the strength of the (Haim et al. 2022) attack when we know a sufficient amount of prior information about the data? If not, does your paper's result move us closer to gaining such guarantees?

References in my review above:
(Haim et al. 2022) - cited in the authors' paper
(Lyu and Li 2020) - cited in the authors' paper
(Ji and Telgarsky 2020) - cited in the authors' paper
Chaudhuri, Kamalika, et al. "Guarantees of confidentiality via Hammersley-Chapman-Robbins bounds." arXiv preprint arXiv:2404.02866 (2024).
Struppek, Lukas, et al. "Plug & play attacks: Towards robust and flexible model inversion attacks." arXiv preprint arXiv:2201.12179 (2022).

---

> ### Author Response · Authors · 2025-11-24
>
> We thank the reviewer for their time and detailed, constructive review. We appreciate the reviewer for highlighting the importance of our theoretical analysis of the role of prior knowledge in reconstruction attacks and for expanding beyond the conditions necessary for the (Haim et al. 2022) attack.
>
> W1. **Data priors in experiments** We thank the reviewer for their suggestion. We added the following discussion to our experiment on the sphere: “We evaluate two types of prior. The first one is the magnitude of the data, and the second one is its geometry. Spheres with different radii have different geometry, such as, for example, different curvature and different typical inner product and distance between two randomly sampled points.”
>
> W2. **Suggesting directions for future work** We added the following discussion in the revised paper in Section 5: “Expanding our findings to more modern, non-homogeneous neural networks is an important and interesting question for future work.”
>
> W3. **Adaptive attacks** Thanks for bringing the adaptive attack perspective to our attention. We added the following discussion in Section 4: “The experiments show a clear link between the attacker’s prior knowledge and attack effectiveness: stronger or more accurate priors yield stronger attacks. Thus, an adaptive adversary who can approximate the true prior gains an advantage. Addressing how such priors can be estimated and how to defend against attackers who do so is left for future work.”
>
> C1. **Adding $\lambda_i$**: The reviewer is correct, and we fixed the stationary loss in (6) to also include $\lambda_i$
>
> C2.**Small radii**  Regarding experiments with radii less than 1, we added the suggested experiment (Figure 3, Appendix F). We added initializations using spheres of radius 0.1 and 0.5. These initializations perform better because, in high-dimensional spaces, points sampled uniformly from the unit sphere (radius 1), even though the training data lies there, tend to be extremely far apart. In contrast, sampling from a smaller-radius sphere produces initialization points that are closer to the training data than points sampled from the full unit sphere.

---

> > ### Author Response · Authors · 2025-11-24
> >
> > Q. **Prior guarantees** Currently, to the best of our knowledge, sufficient conditions on the prior distribution required for the successful application of the Haim et al. (2022) attack remain an open problem. Our paper provides a foundational result by demonstrating that prior knowledge is a necessary condition for the attack to succeed. This establishes a critical lower bound and shifts the research focus toward the inverse problem of identifying the specific sufficient conditions that guarantee success.

---

### Official Review · Reviewer_73hu · 2025-11-01

**Soundness:** 3
**Presentation:** 3
**Contribution:** 2
**Rating:** 4
**Confidence:** 3

**Summary:**

The paper analyzes why implicit-bias–based reconstruction attacks on trained neural networks can fail, arguing that without explicit priors about the data domain, the KKT-driven objective used by prior attacks admits many alternative optima far from the true training set. The paper formalizes constructive merge/split operations that generate indistinguishable KKT sets, extends the analysis to approximate KKT points, and presents experiments on synthetic data and CIFAR, indicating that weaker priors degrade reconstructions while better-optimized networks are less susceptible, thereby reframing when leakage is feasible.

**Strengths:**

- This paper offers a clear formalization of the attack objective and isolates the role of priors by analyzing the KKT-only loss.
- This paper offers practical guidance to related fields by indicating when reconstruction is unlikely without priors, which helps motivate practical countermeasures.
- The writing is clear.

**Weaknesses:**

- The theory is developed for binary classification with homogeneous ReLU models, and it's not clear which proof steps would still hold in multiclass or with other losses. A brief *"what carries over vs. what doesn't"* discussion, or a tiny multiclass sanity check, would help readers judge how far the results travel.

- Even if the Discussion later clarifies that results pertain to KKT-based reconstruction without priors, the abstract/Section 1 do not foreground this scope where readers first encounter the contributions; briefly surfacing the scope at the outset would align expectations and reduce the risk that practice-facing takeaways are read as general privacy claims, especially given prior work that studies broader regimes (multiclass, other losses) [1, 2].

- The proposed shift/bias trick is intuitive but lacks robustness guarantees and could be learned by an adaptive adversary; scoping the claim (e.g., "effective for KKT-only objectives under unknown shifts") would avoid overgeneralization.

- The discussion suggests implicit bias may hinder leakage along the KKT route, but memorization and extraction remain documented in overparameterized and generative settings [3, 4]. The authors should make it clearer: which leakage channels your analysis does and does not cover, so readers don’t overgeneralize the safety implications.

## References

- [1] N. Haim et al. Reconstructing Training Data from Trained Neural Networks. NeurIPS 2022.
- [2] G. Buzaglo et al. Deconstructing Data Reconstruction: Multiclass, Weight Decay and General Losses. NeurIPS 2023.
- [3] V. Feldman and C. Zhang. What Neural Networks Memorize and Why: Discovering the Long Tail via Influence Estimation. NeurIPS 2020.
- [4] N. Carlini et al. Extracting Training Data from Large Language Models. USENIX Security 2021.

**Questions:**

- Beyond explicitly removing domain priors in the objective, which elements of the training pipeline might still act as priors (data normalization, augmentations, early stopping, weight decay), and how did you control or record them so readers can interpret the reconstruction outcomes?
- Did you try residual connections to see if the phenomena qualitatively persist?

---

> ### Author Response · Authors · 2025-11-24
>
> We thank the reviewer for their careful and constructive review and appreciate the positive feedback on the practical guidance our results offer for avoiding data reconstruction, as well as on our clear formalization of the KKT-based reconstruction objective and the role of priors. We address your concerns below.
>
> W1. **What carries over:** Although the attack in [1] is originally formulated for binary classification and assumes that the trained network has converged to a KKT point, [2] extends it to a multiclass attack by using the corresponding multiclass KKT conditions. Since this generalized attack in [2], like the one in [1], is also based on the geometric relationship between the weight space and the training set, our analysis generalizes directly, and the main conclusions remain unchanged.
>
> To address the reviewer’s request, we additionally performed an experiment in the multiclass setting (Figure 8, Appendix F). We trained two multiclass classifiers on CIFAR-10 with a convolutional neural network (CNN) as the first layer: one on images shifted by 0.01 and another on images shifted by 0.1. We then attacked these models using the multiclass KKT-based method of [2], i.e., by minimizing the multiclass KKT loss. The results show that the reconstruction quality of the network trained with a shift of 0.01 is higher than that of the network trained with a shift of 0.1, empirically supporting our claim that, in the multiclass setting, a sufficiently strong prior is necessary to reconstruct the training examples.
>
> W2. **Scoping the discussion:** Our intention was already to scope the work to the KKT-based framework of Haim et al. and to the corresponding reconstruction setting, and we attempted to make this explicit in the introduction, as it (i) anchors our analysis specifically in the KKT/implicit-bias–based attack of Haim et al. and (ii) indicates that our results are about this particular reconstruction mechanism.
> In the revised version of the manuscript, we have clarified the scope of our analysis in the introduction by adding a suitable footnote.
>
> W3. **Adaptive attacker:** We appreciate the reviewer’s concern about robustness and the possibility that an adaptive adversary could learn the shift. To address this, we added an additional experiment in which there is no single global shift that could be inferred and undone by a sophisticated attacker.
>
> **Experiment with a random radius.**
> We constructed a dataset whose points lie in an annulus: all examples have norms between radii 1 and 4 (chosen arbitrarily) and are drawn from this ring (support). In particular, the data do not concentrate on a single radius that could be “decoded” as a simple shift parameter, as in the scenario raised by the reviewer. We then ran the reconstruction attack multiple times with different initializations (priors):
> Initialization with radii between 1 and 4 (matching the support of the true distribution),
>
> Initialization with radii between 2 and 5 (partially overlapping support).
>
> Initialization with radii between 6 and 9 (no overlap with the true support).
> We measured the average distance of the 5  best reconstructions from the original training set.
>
> As shown in the new experimental results in Appendix F of the revised version, reconstruction quality improves as the prior’s support overlaps more with the true data distribution. In particular, when there is no overlap (case 3), meaningful recovery does not occur.
>
> These results support our claim that, in the KKT-based setting, successful reconstruction effectively requires a prior that overlaps the true data distribution, and they make precise the sense in which our “shift/bias” idea should be interpreted and scoped.
>
> W4. **Memorization vs. leakage** A network can memorize its training data in a strong sense, yet we may still lack a principled or efficient method to reconstruct that data from its parameters.  In other words, memorization alone does not automatically imply leakage: without a concrete attack mechanism, the memorized information may remain practically inaccessible.  The KKT-based attacks we study provide a concrete mechanism for extracting information from a trained model, and our work investigates the limitations of this mechanism.
> We now make this scope more explicit in the discussion: our results characterize leakage along the KKT-based reconstruction channel (as in [1, 2]), providing a detailed understanding of this important class of attacks. While our focus is not on other leakage mechanisms, such as those studied in overparameterized or generative settings [3, 4], we view our analysis as complementary to that line of work.

---

> > ### Author Response · Authors · 2025-11-24
> >
> > Q1. **Priors imposed by training** We agree that elements of the training pipeline can implicitly act as priors. We clear which components were used and how they affect our conclusions.
> > Normalization. Input normalization indeed functions as an implicit prior, as it encodes prior knowledge that the data are centered (e.g., mean zero). Precisely because it introduces such information, we chose not to use input normalization in our experiments.
> > Augmentations. Empirically, we find that augmented examples are rarely recovered (in most runs, almost never). We believe this is because these augmented samples typically do not belong to the set of margin points, which are precisely the examples most susceptible to recovery. For the small subset of augmented examples that are recovered, i.e., those for which the net output value indicates the margin, the recovery rate is comparable to that of original margin examples. Overall, when we include augmentations in the training set, the fraction of recovered examples remains similar, and we do not observe any noticeable change in the attack's convergence rate or in the quality of the recovered examples.
> > Early stopping. In principle, early stopping could interfere with convergence to the KKT point. To check this, we repeated our experiments with a standard early-stopping criterion (training stops when there is no improvement on a validation metric). We did not observe qualitative differences in the reconstruction behavior under these settings.
> > Weight decay. In practice, small weight-decay values primarily affect the time required for successful reconstruction (reconstruction time), whereas large values substantially degrade the ability to reconstruct examples. Theoretically, with weight decay, the stationarity of the regularized loss yields (cf. Eq.~(15) [2]),
> >     $\theta \approx \sum_{i} \lambda_i \,\nabla_\theta \Phi(x_i; \theta),$
> > which has exactly the same KKT form as in the basic reconstruction attack. Consequently, once training drives the regularized gradient close to zero, this relation approximately holds, and the stationarity-KKT loss/objective can be directly applied.
> > WD only changes the KKT-style stationarity relation (e.g., Eq.(15) [2]) that the attack exploits, without injecting additional dataset/domain knowledge into the reconstruction objective. Thus, is not a prior.
> > In our reported experiments, we avoid weight decay.
> >
> > Q2. **Residual Networks** Empirically, we repeat the experimental setting (“Beyond the theoretical framework”) in section 4 with two and three ResNet classification blocks on CIFAR-10. When attacking the network, we were unable to reconstruct any image. This is because the network did not converge to a KKT point during training. To the best of our knowledge, no other prior work has successfully utilized such similar implicit bias attacks on ResNet architectures.

---

### Meta-Review · Area_Chair_d4hQ · 2026-01-02

**Summary:**

This paper investigates the limitations of reconstruction attacks of the training dataset based on a KKT objective as introduced by [Haim et al., 2022]. The authors show that such objective admits optima that are arbitrarily far from the true training set. Thus, merging/splitting techniques lead to alternative training sets that are far from the underlying correct training data but that still minimize the objective of the reconstruction attack. Numerical experiments corroborate the validity of the theoretical findings demonstrating the failure modes of the attack in a variety of settings.

While the 5 reviewers appreciated the novelty of the idea, there was quite a mix in the scores and several concerns were raised. The authors provided a thoughtful rebuttal, with several new experiments which addressed most of such concerns. Therefore, I am happy to recommend acceptance.

**Reviewer Concerns:**

Reviewer 73hu. The concern about the multi-class setting was addressed in a satisfactory manner. I also appreciated the discussion about the priors imposed by training and experiments where points lie on an annulus. Thus, my opinion is that most of the concerns have been addressed.

The response to Reviewers e1G7 and 3CEM, who were already positive, was satisfactory.

Reviewer 2UcW. The authors added multiple experiments (MNIST, annulus, ImageNet, multi-class) and discussed in a convincing way the KKT loss gap, as well as the KKT loss in the original attack. Thus, my opinion is that most of the concerns have been addressed.

Reviewer 1FR8 raised concerns about limited empirical validation and lack of quantitative evaluation of prior strength, which were resolved by the rebuttal.

**Reviewer Scores:**

While it is hard to say for sure, I find it likely that a full discussion could have led to consensus towards accepting the paper.

---

### Decision · Program_Chairs · 2026-01-26

Accept (Poster)